# Efficient event-based delay learning in spiking neural networks

Balázs Mészáros [1,2] ✉, James C. Knight [1] & Thomas Nowotny [1] ✉

Spiking Neural Networks compute using sparse communication and are attracting increased attention as a more energy-efficient alternative to traditional Artificial Neural Networks. While standard Artificial Neural Networks are stateless, spiking neurons are stateful and hence intrinsically recurrent, making them well-suited for spatio-temporal tasks. However, the duration of this intrinsic memory is limited by synaptic and membrane time constants. Delays are a powerful additional mechanism and, in this paper, we propose an event-based training method for Spiking Neural Networks with delays, grounded in the EventProp formalism, which enables the calculation of exact gradients with respect to weights and delays. Our method supports multiple spikes per neuron and introduces a delay learning algorithm that can, in contrast to previous methods, also be applied to recurrent Spiking Neural Networks. We evaluate our method on a simple sequence detection task, as well as the Yin-Yang, Spiking Heidelberg Digits, Spiking Speech Commands and Braille letter reading datasets, demonstrating that our algorithm can optimise delays from suboptimal initial conditions and enhance classification accuracy compared to architectures without delays. We also find that recurrent delays are particularly beneficial in small networks. Finally, we show that our approach uses less than half the memory of the current state-of-the-art delay-learning method and is up to 26 × faster.

Artificial Neural Networks (ANNs) have gained immense popularity and seen significant improvements over the past decade. However, commonly used models are very energy intensive[1], whereas the human brain has an estimated power budget of only 20 W[2]. It might, therefore, be beneficial to again turn to neuroscience for inspiration. One reason for the brain's better efficiency is that neurons in the brain transmit sparse binary events called spikes, while the units in ANNs typically communicate real-valued activation values densely in time and space. Spiking Neural Networks (SNNs) leverage the sparse communication patterns of biological neurons for machine learning (ML) and are particularly efficient on neuromorphic systems designed to provide efficient hardware platforms for brain-like computing[3–5]. Like ANNs, SNNs are universal function approximators, suggesting they could enable an energy-efficient future for ML. Therefore, researchers are

increasingly focusing on training SNNs on popular ML tasks, for instance, in the fields of computer vision[6] and natural language processing[7]. Since individual spiking neurons rely on hidden temporal dynamics, they have 'implicit recurrence' so, even SNNs with feedforward architectures, have theoretical advantages for temporal processing over feedforward networks of stateless units. However, despite this potential, achieving or beating the performance of ANNs using SNNs remains a significant challenge.

The most commonly used training algorithm in ANNs is gradient descent. However, the non-differentiable transitions at spike times cause mathematical complications when calculating gradients in SNNs. To overcome this issue, some researchers do not train SNNs directly but instead train ANNs and then transfer the weights to an SNN for inference[8,9]. However, because this approach typically uses spike

[1]Sussex AI, School of Engineering and Informatics, University of Sussex, Brighton, United Kingdom. [2]The Alan Turing Institute, London, United Kingdom. ✉e-mail: b.mszros@sussex.ac.uk; t.nowotny@sussex.ac.uk

counts to represent the activations of ANN units, it does not fully leverage the potential energy savings of sparse spiking in SNNs. While there are more efficient encoding alternatives[10], one time step in the ANN is still mapped to many timesteps in the SNN and the efficiency of SNNs is not leveraged at training time.

Another popular solution is to discretise the network dynamics and use Backpropagation Through Time (BPTT) with 'surrogate gradients' to smooth the threshold function in the backward pass[11,12]. However, this method requires storing neuron state variables at every time step for the backward pass, meaning that memory requirements scale linearly with sequence length. This limits the maximum number of time steps in a trial to a few hundred. Furthermore, this method also does not exploit the increased efficiency of sparse spiking during the backward pass.

Bohte et al.[11] were the first to show how to calculate exact gradients in SNNs, by providing recursive relations for the gradient that can be implicitly computed using backpropagation. Alternatively, with some constraints on the time constants of neurons, analytic expressions for the time of the next spike of a leaky integrate-and-fire (LIF) can be derived and differentiated[13,14], enabling the calculation of gradients. However, more generally, neurons in SNNs exhibit hybrid dynamics (a longstanding focus in optimal control theory[15]), combining continuous changes between spikes with discontinuous state transitions at spike times. The link between neural network training and optimal control has been well established[16], and the adjoint method – a staple of optimal control – has been used to derive gradients for smoothed spiking neuron models without reset[17]. Also using the adjoint method, Wunderlich and Pehle[18] developed the EventProp algorithm for calculating exact gradients in SNNs of integrate-and-fire neurons and 'exponential synapses'. The backward pass in EventProp combines a system of ordinary differential equations for the adjoint variables of the neuron dynamics with purely event-based backward transmission of error signals at spike times, making the best use of the hybrid nature of SNNs. Wunderlich and Pehle tested their method on latency-encoded MNIST[19] and the Yin-Yang datasets[20]. More recently, Nowotny et al.[21] extended EventProp to the more challenging benchmarks of the Spiking Heidelberg Digits (SHD) and Spiking Speech Commands[22], using loss shaping to overcome issues caused by exact gradients not containing information about spike creation and deletion. The time and space complexity of EventProp enables very efficient GPU training of large models on long sequences[21], hardware-in-the-loop training[23], and even training on neuromorphic hardware[24].

Spiking neurons' implicit recurrence is characterised by temporal parameters such as the membrane and synaptic time constants. Research has shown that optimising these[25,26] can enhance network performance. Delays are another mechanism for temporal processing, and recent work indicates the utility of learnable delays for temporal tasks[27,28], In biological neural networks, synaptic delays arise naturally due to the spatial structure of the network and can be modified to facilitate coincidence detection[29] and learning[30]. From a computational perspective, the inclusion of delays has been shown to significantly increase network capacity[31] and Maass and Schmitt[32] demonstrated that an SNN with $k$ adjustable delays can compute a much richer class of functions than a network with $k$ adjustable weights. Furthermore, neuromorphic systems such as SpiNNaker[33] and Loihi[34] are specifically designed to accommodate synaptic delays, so that SNNs with delays can still be efficiently deployed. Adding delays to SNNs has recently gained popularity, with several models treating them as learnable parameters and obtaining state-of-the-art performance on classification tasks[27,28,35]. Grappolini and Subramoney[36] even showed that networks can be trained to comparable performance with pure synaptic delay learning. However, most of these methods are based on surrogate gradients[27,28,37], which do not allow the event-based nature of SNNs to be exploited at training time, and some use temporal convolutions to implement delays[27,38], which results in large overheads

in memory and computation. DelGrad[39] was the first exact gradient-based delay learning method, but so far it has only been implemented in feedforward networks where each neuron only emits one spike per trial. No prior work has implemented delay learning for recurrent connections.

Here, we extend EventProp to incorporate heterogeneous and learnable delays and implement our extended version in mlGeNN[40,41] – a spike-based ML library built on the GPU-optimized GeNN simulator[42–44]. GeNN generates GPU kernels for efficiently simulating networks of neurons which communicate with sparse events using a *hybrid* simulation strategy where the forward and backward dynamics of each neuron are updated every timestep, but synapses are only updated in timesteps where their presynaptic neurons produce a spike. This hybrid strategy differs from the purely time-driven approach (typically used to implement SNNs with standard ML libraries), where neurons and synapses are updated every timestep and also from purely event-based simulators where both neurons and synapses are only updated at spike times. All three approaches have advantages and disadvantages. While a purely timestep-based approach is inherently wasteful, standard ML libraries counteract the inherent inefficiency with highly GPU-optimised matrix multiplication routines to multiply neuron outputs with weights. Pure event-based simulators make efficient use of the event-based nature of SNNs and produce precise spike times, unconstrained by a timestep grid, but this comes at a cost. Only a limited subset of neuron models[45,46] have dynamics that can be directly interpolated between events, algorithms and data structures become increasingly complicated if recurrent connectivity and delays are required[47] and, the perceived computational advantage of event-based simulation dwindles rapidly as the frequency of events increases[48]. For example, each input in the SHD dataset fires at ~10 spikes per second. If we consider a hidden neuron densely connected to these inputs, it receives 7000 spikes per second, meaning that an event-based neuron would have to update 7× more frequently than one simulated using a 1 ms timestep.

Nowotny et al.[21] described the initial GeNN implementation of EventProp, which has subsequently been implemented as a 'compiler'[41] for mlGeNN. Here, we have added delays to the mlGeNN EventProp compiler, enabling the easy exploration of delay learning in a wide range of network architectures. When using our method, increasing the range of delays only requires enlarging a *per-neuron* buffer, resulting in a much lower memory overhead than convolution-based approaches and thus allowing efficient handling of long delays. Our method further allows the outputs of neurons to feed into general spike- and/or voltage-dependent loss functions, offering great flexibility in designing training objectives. Our approach outperforms prior work using EventProp on the SHD and SSC datasets[21] – achieving superior performance with almost 5× fewer parameters. Additionally, we demonstrate a speedup of up to 26× and memory savings of over 2× compared to surrogate-gradient-based dilated convolutions implemented in PyTorch[27].

## Results

The EventProp algorithm is an application of the adjoint method for calculating sensitivities in hybrid dynamical systems[49,50] to the problem of calculating the gradient of a loss function in an SNN. From an ML perspective, it can be considered an event-driven form of the popular BPTT gradient descent-based learning in RNNs. The difference is that the gradient computation employs a hybrid approach: the derivatives are determined through both continuous differential equations and discrete state transitions of adjoint variables that occur at saved spike times.

Intuitively, in an SNN, synaptic weights can only affect the network activity, and hence cause loss, at times when a spike is transmitted. The adjoint variables track the potential loss caused by each neuron, and their value at the time of a spike quantifies how much loss that spike

contributed to the overall loss, essentially assigning responsibility or "blame" to both the spike event and the synaptic weights involved. For output neurons, blame for loss is directly added to the adjoint variable

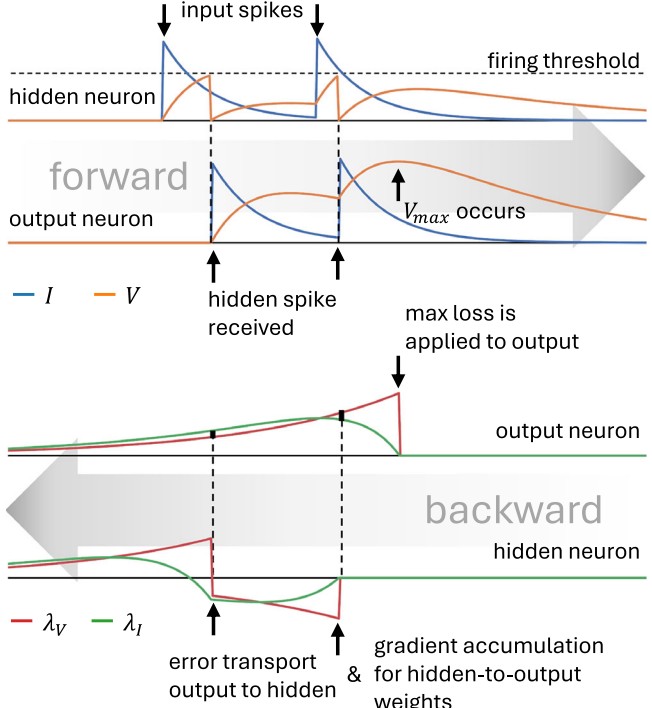

**Fig. 1 | Illustration of the original EventProp formalism without delays.** In a minimal example, a network has input neurons, one hidden layer and an output layer. Input spikes cause instantaneous jumps in the hidden **I** variable (blue lines), which drives the **V** variable (orange lines). When **V** reaches the firing threshold, it is reset and spikes are emitted, which instantaneously jump the **I** variable of the output neurons. This forward pass is followed by a backward pass, where the "blame" of each weight for the eventual loss is calculated. The adjoint variables $\boldsymbol{\lambda}_V$ and $\boldsymbol{\lambda}_I$ are proxies of this blame. The calculation occurs backwards in time. Here we illustrate the use of a readout and loss that are based on the maximum voltage of the output neurons. Accordingly, the loss causes a jump in the output $\boldsymbol{\lambda}_V$ variable at the time when the maximum output voltage occurred in the forward pass. This blame is then transported as jumps of $\boldsymbol{\lambda}_V$ of hidden neurons at the times when hidden spikes had occurred. Gradient updates to the hidden-to-output weights occur at the same time. Note that the plots are for illustrative purposes only and not to scale, other than matching pairs of $\boldsymbol{\lambda}_V$ and $\boldsymbol{\lambda}_I$ being at the same scale.

corresponding to the membrane dynamics (jump in $\boldsymbol{\lambda}_V$ of the output neuron where $V_{\max}$ occurred in Fig. 1), while for internal neurons, it propagates backward from downstream connections (jumps in $\boldsymbol{\lambda}_V$ of the hidden neuron at saved spike times in Fig. 1). The gradient for each synaptic weight ($w_{ji}$) accumulates across pre-synaptic firing events, capturing the moments when that connection influenced the post-synaptic neuron's behaviour and consequently affected network performance. The differential equations governing the adjoint variables precisely track how input fluctuations drive membrane changes that ultimately impact the loss – whether by triggering spikes or directly affecting the output neuron's contribution to the loss function.

Here, we derive an extended EventProp formalism that extends the original algorithm in both, that it can be applied to SNNs with delays and that it enables learning of suitable delays, i.e. to calculate gradients of a loss function with respect to delays. Although the EventProp formalism accommodates various neuron models[51], for simplicity, we will describe it for the LIF neurons and exponential current-based synapses employed by Wunderlich and Pehle[18].

The forward pass of the SNN is described by first-order ordinary differential equations for the dynamics of the current **I** and voltage **V**; and discontinuous jumps in the variables at the occurrence of spikes (see Table 1). Note that this only differs from the forward pass described by Wunderlich and Pehle[18] due to the delay $d_{mn}$ between neuron $n$ and neuron $m$ which causes the $k$-th jump in the network in the current $I_m$ of neuron $m$ caused by a spike in neuron $n$ at time $t_k^{\mathrm{spike}}$ to occur at time $t_k^{\mathrm{spike}} + d_{mn}$, see also Fig. 2, top. If all $d_{mn}$ are zero, this reverts back to the original EventProp formalism.

To obtain the backward pass for our network with delays, we take the derivative of the loss function with respect to a weight, $w_{ji}$. The loss function can depend directly on spike timing (which is compatible with the gradient calculation because LIF neuron spike times vary smoothly with weight changes), or it can be expressed as an integral over the voltage **V** (where the integral effectively smooths out the discontinuities that would otherwise make the gradient calculation difficult when spikes occur).

$$\frac{\mathrm{d}\mathscr{L}}{\mathrm{d}w_{ji}} = \frac{\mathrm{d}}{\mathrm{d}w_{ji}}\left[l_p(\mathscr{S}) + \int_{t=0}^{T} l_V(\mathbf{V}, t)\mathrm{d}t\right], \quad (1)$$

where $l_p$ is the loss term that depends on the set of output spike times $\mathscr{S} \equiv \{t_k^{\mathrm{spike}} \mid k = 1, \ldots, N_{\mathrm{spike}}\}$, and $l_V$ is the voltage-dependent loss. Following the adjoint method, we then add Lagrange multipliers $\boldsymbol{\lambda}_I$ and $\boldsymbol{\lambda}_V$, multiplied by the continuous dynamics, which can be interpreted as

**Table 1 | Forward and backward propagation of a Leaky Integrate-and-Fire (LIF) neuron**

| Free dynamics | Transition condition | Jumps at transition |
|---|---|---|
| Forward: | | |
| $\tau_m \dot{\mathbf{V}} = -\mathbf{V} + \mathbf{I}$ | $V_{n(k)}^- - \vartheta = 0,$ | $V_{n(k)}^+\big|_{t_k^{\mathrm{spike}}} = 0$ |
| $\tau_s \dot{\mathbf{I}} = -\mathbf{I}$ | $\dot{V}_{n(k)}^- \neq 0$ | $I_m^+\big|_{t_k^{\mathrm{spike}}+d_{mn(k)}} = I_m^-\big|_{t_k^{\mathrm{spike}}+d_{mn(k)}} + w_{mn(k)}$ |
| Backward: | | |
| $\tau_m \boldsymbol{\lambda}_V' = -\boldsymbol{\lambda}_V - \frac{\partial l_V}{\partial \mathbf{V}},$ | $t - t_k^{\mathrm{spike}} = 0,$ | $\lambda_{V,n(k)}^- = \left[\frac{\dot{V}_{n(k)}^+}{\dot{V}_{n(k)}^-}\lambda_{V,n(k)}^+ + \frac{1}{\tau_m \dot{V}_{n(k)}^-}\left[\frac{\partial l_p}{\partial t_k^{\mathrm{spike}}} + l_V^- - l_V^+\right]\right]\Big|_{t_k^{\mathrm{spike}}}$ |
| $\tau_s \boldsymbol{\lambda}_I' = -\boldsymbol{\lambda}_I + \boldsymbol{\lambda}_V$ | for any $k$ | $+ \left[\frac{1}{\tau_m \dot{V}_{n(k)}^-}\right]\Big|_{t_k^{\mathrm{spike}}} \sum_m w_{mn(k)}\left[\lambda_{V,m}^+ - \lambda_{I,m}^+\right]\big|_{t_k^{\mathrm{spike}}+d_{mn(k)}}$ |

| **Gradient updates** | |
|---|---|
| Weight learning: | Delay learning: |
| $\frac{\mathrm{d}\mathscr{L}}{\mathrm{d}w_{ji}} = -\tau_s \sum_{\left\{t_k^{\mathrm{spike}} \mid n(k)=i\right\}} \lambda_{I,j}\big|_{t_k^{\mathrm{spike}}+d_{ji}}$ | $\frac{\mathrm{d}\mathscr{L}}{\mathrm{d}d_{ji}} = -w_{ji} \sum_{\left\{t_k^{\mathrm{spike}} \mid n(k)=i\right\}} \lambda_{I,j} - \lambda_{V,j}\big|_{t_k^{\mathrm{spike}}+d_{ji}}$ |

The vectors **V** and **I** are the membrane potentials and input currents and $\boldsymbol{\lambda}_V$ and $\boldsymbol{\lambda}_I$ the corresponding adjoint variables. $\tau_m$ and $\tau_s$ are the membrane and synaptic time constants. $w_{mn}$ is the synaptic weight and $d_{mn}$ is the synaptic delay from neuron $n$ to neuron $m$. $\vartheta$ is the firing threshold. The dot denotes the derivative with respect to time, and the prime the derivative backwards in time. Superscript "−" and "+" denote the values before and after a transition. $t_k^{\mathrm{spike}}$ denotes the $k^{\mathrm{th}}$ spike and $n(k)$ the index of the neuron that fired this spike. $l_p$ and $l_V$ define the shape of the loss function. Without the transition condition (in other words, with $\vartheta = \infty$), we arrive at the Leaky Integrator (LI) neuron. We highlight all our additions in red.

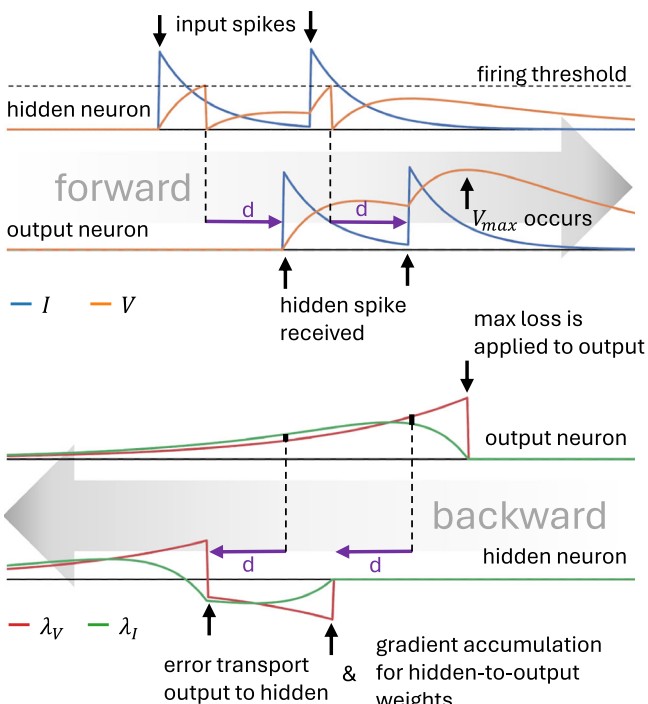

**Fig. 2 | Illustration of the extended EventProp algorithm for SNNs with delays.** In essence, the forward pass works in the same way as for networks without delays (Fig. 1), except that the jump in post-synaptic **I** variables occurs with a delay $d$. In the backward pass, this translates into backwards transport of blame at the original spike times but transporting values of post-synaptic adjoint variables at a delayed time (in forward time). Gradients with respect to weights and delays are accumulated at the same saved spike times and based on the same delayed quantities.

dynamic constraints,

$$\frac{\mathrm{d}\mathscr{L}}{\mathrm{d}w_{ji}} = \frac{\mathrm{d}}{\mathrm{d}w_{ji}}\left[l_p(\mathscr{S}) + \int_{t=0}^{T}\left[l_V(\mathbf{V},t) + \boldsymbol{\lambda}_V\mathbf{f}_V + \boldsymbol{\lambda}_I\mathbf{f}_I\right]\mathrm{d}t\right], \quad (2)$$

where $\mathbf{f}_V \equiv \tau_\mathrm{m}\dot{\mathbf{V}} + \mathbf{V} - \mathbf{I}$ encodes the voltage dynamics constraint and $\mathbf{f}_I \equiv \tau_\mathrm{s}\dot{\mathbf{I}} + \mathbf{I}$ the current dynamics constraint.

The essence of the original EventProp derivation was to split the integral in (2) at the $N_\mathrm{spike}$ spike times $t_k^\mathrm{spike}$ when the jumps occur. Between the jumps, everything is well-defined and the standard adjoint method is easily applied – resulting in the backward dynamics for the adjoint variables. With some work (see derivations by Wunderlich and Pehle[18]), the values of adjoint variables before and after jump times in the backward pass can then be defined so that the remaining expression for the gradient becomes a simple sum over $\boldsymbol{\lambda}_I$ values at spike times, leading to an event-based weight update rule:

$$\frac{\mathrm{d}\mathscr{L}}{\mathrm{d}w_{ji}} = -\tau_\mathrm{s}\sum_{\{t_k^\mathrm{spike}\,|n(k)=i\}}\lambda_{I,j}|_{t_k^\mathrm{spike}}. \quad (3)$$

We apply a similar approach here, but in our network with delays, spike emission and arrival times become separate events. We address this by extending the set of spike times to the set of all event times that include both spike emission times $t_k^\mathrm{spike}$ and spike arrival times $t_k^\mathrm{spike} + d_{mn(k)}$, where $n(k)$ is the index of the neuron that fired the $k^\mathrm{th}$ spike:

$$\mathscr{E} \equiv \mathscr{S} \cup \{t_k^\mathrm{spike} + d_{m,n(k)}\,|\,k=1,\dots,N_\mathrm{spike}, m=1,\dots,N\}. \quad (4)$$

We denote the elements of this set as $t_k^\mathrm{event} \in \mathscr{E}$ and assume that they are ordered such that $t_k^\mathrm{event} \leq t_{k\prime}^\mathrm{event}$ for $k < k\prime$. We can then proceed in the same way as in the original EventProp derivation. The resulting

backward pass becomes computable because all delays are non-negative, meaning that by the time we compute the adjoint variables for the spiking neuron $i$ at time $t$, the adjoint variables for the post-synaptic neurons $j$ receiving the spike at time $t + d_{ji}$ will have already been calculated before, in backward time (see Fig. 2, bottom).

After extensive calculations (Section "Methods"), we arrive at a formula that remains fully event-based for synaptic actions in the backward pass and thus can be efficiently computed,

$$\frac{\mathrm{d}\mathscr{L}}{\mathrm{d}w_{ji}} = -\tau_\mathrm{s}\sum_{\{t_k^\mathrm{spike}\,|n(k)=i\}}\lambda_{I,j}|_{t_k^\mathrm{spike} + d_{ji}}. \quad (5)$$

Using the same approach, we can also derive the derivative of the loss function with respect to the delays $d_{ji}$. Remarkably, the derived backward dynamics of the adjoint variables remain the same, meaning that using the exact same $\boldsymbol{\lambda}_I$ and $\boldsymbol{\lambda}_V$ dynamics in the backward pass, we *can also perform gradient descent on synaptic delays in an event-based manner*. The resulting formula for the delay gradients is

$$\frac{\mathrm{d}\mathscr{L}}{\mathrm{d}d_{ji}} = -w_{ji}\sum_{\{t_k^\mathrm{spike}\,|n(k)=i\}}(\lambda_{I,j} - \lambda_{V,j})|_{t_k^\mathrm{spike} + d_{ji}}. \quad (6)$$

We thus end up with an event-based weight and delay learning algorithm, which enables delay learning in networks with multiple recurrently-connected layers and with multiple spikes per neuron.

## Sequence detection task

To test the effectiveness of our method, we evaluated it in various machine learning settings. First, we generated a simple binary classification task that can be solved with perfect accuracy using optimal delays. Specifically, we used two LIF neurons connected to two LI neurons (see Table 1), with all connection strengths set to 1. The task involves two classes:

- Class 1: The first input neuron emits a spike at 0 ms, and the second emits a spike at 10 ms.
- Class 2: The first input neuron emits a spike at 10 ms, and the second emits a spike at 0 ms.

The output of the network is determined based on the maximum voltage reached by the two output neurons, each corresponding to one of the output classes. The class associated with the neuron that reaches the higher voltage is selected as the network's prediction. We can observe that having 10 in the diagonals and 0 everywhere else in our 2 × 2 delay matrix solves the task. We started with the least optimal delay distribution – delays on the diagonals being 0, and 10 everywhere else – and, with the learning rate set to 1, we achieve 100% accuracy after encountering both examples 6 times. This result demonstrates that, by introducing our delay updates into the learning framework, SNNs become capable of not only coincidence but sequence detection. Figure 3 illustrates the gradient calculation in this task.

## Yin-Yang dataset

We also experimented with the Yin-Yang dataset[20], which has been tested using both EventProp (without delays)[18] and DelGrad[39], see Fig. 4. Similarly to DelGrad, we looked at feedforward networks and varied the size of the hidden layer from 5 to 30. We initialised all delays at 0, allowing them to evolve in the range of 0–10 during training. DelGrad also applies a sigmoid on the delays, but in our experiments we deemed it unnecessary. Otherwise, our implementation is very close to DelGrad – we implemented a similar weight-bumping mechanism, and allowed neurons to only spike once by increasing the refractory period to a value higher than the sequence length. We also trained using the time-invariant mean squared error loss[39], see section "Methods" for

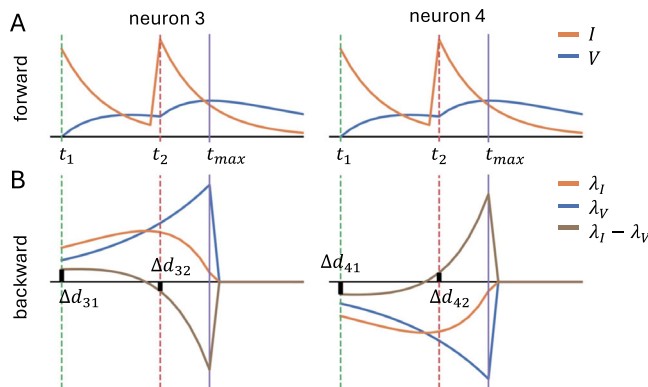

**Fig. 3 | Illustration of the learning updates in the sequence detection task.**
**A** Voltage **V** and current **I** in the forward pass. **B** Adjoint variables $\lambda_V$ and $\lambda_I$. The loss is injected into $\lambda_V$ at $t_{\max}$ when the output voltage reached maximum in the forward pass. This then propagates into $\lambda_I$ and eventually leads to delay updates at the saved spike times $t_1$, $t_2$. For neuron 3, the update to $d_{32}$ at $t_2$ is negative and the update to $d_{3,1}$ at $t_1$ positive, meaning that (subject to delays being non-negative) the excitatory postsynaptic potential (EPSP) from neuron 2 is moved to earlier and the EPSP from neuron 1 to later, moving them close together. For neuron 4, the opposite is the case, where the updates are such that the EPSPs are moved apart. This is exactly what is needed to increase the maximal output voltage of neuron 3 and decrease the maximal output of neuron 4. When the other input class is active, all roles are inverted, again leading to the correct delay updates.

derivation. Our findings are similar to DelGrad – with a fixed number of parameters, networks perform similarly (i.e. halving the number of hidden neurons does not decrease performance if delays are introduced). If the number of parameters is not a constraining factor, training delays *and* weights is always advantageous. We achieve the same performance, which is expected, given the equivalence of the gradients[52].

## Spiking Heidelberg digits

Nowotny, Turner, and Knight[21] achieved state-of-the-art results on SHD with a 'delay line' approach, which involved creating 10 copies of the input and cumulatively delaying each by 30 ms. While this architecture achieved high accuracy, it required a large number of parameters, so we instead experimented with learnt delays in the input-to-hidden and hidden-to-hidden connections. For controlling the training dynamics in the hidden population, a target firing rate needs to be set, with the corresponding spike regularisation strength. We kept our target firing rate fixed at 14 spikes per example and treated the regularisation strength as a tunable hyperparameter, which we optimised using 10-fold cross-validation, leaving one speaker out of the training set in each fold. Tuning this parameter was crucial (particularly for networks with recurrent connectivity) and once we identified the best-performing model in cross-validation, we retrained it with the same parameters on the full training set. We enforced early stopping if training accuracy did not improve for 15 epochs. Using this methodology, our best model achieved a training accuracy of 98.47 ± 0.4% and a test accuracy of 93.24 ± 1.0% as depicted in Fig. 5. This configuration included 512

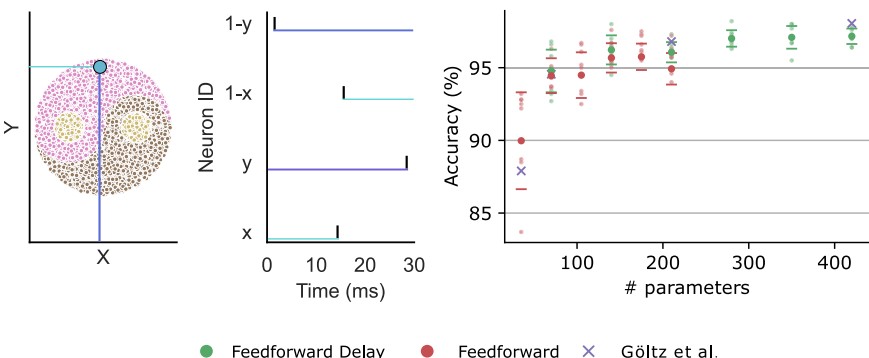

**Fig. 4 | Yin-Yang results.** Left: The Yin-Yang (YY) dataset, with temporal encoding of example datapoint highlighted by a blue dot. Right: We generate separate training, validation, and test sets with 5000, 1000 and 1000 examples respectively; and report the test performance using the model which performed best on the validation set. We look at feedforward networks with and without delays. The purple crosses shows the results reported by Göltz et al.[39]. The points show average accuracy, and the error bars show standard deviation over 8 runs. We also show all individual results as smaller data points.

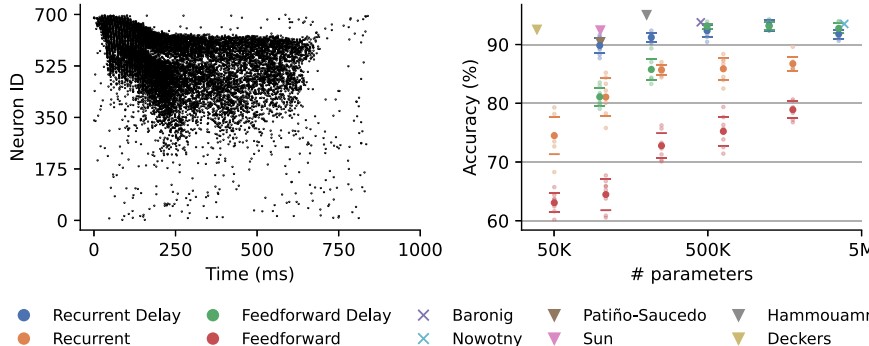

**Fig. 5 | Spiking Heidelberg Digits results.** Left: An example of a speaker saying "five" from the Spiking Heidelberg Digits (SHD) dataset. Right: The SHD dataset does not have a validation set; we perform early stopping when training accuracy does not improve for 15 epochs and report the corresponding test accuracy. Feedforward networks have 2 hidden layers, and recurrent networks have one recurrently connected hidden layer. We implemented models with 128, 256, 512 and 1024 hidden neurons. We also show SOTA results by Baronig et al.[55], and previous EventProp results by Nowotny, Turner, and Knight[21]. The triangles show results of other delay learning methods that appear to have used the test set for validation[27,28,35]. The points show average accuracy, and the error bars show standard deviation over 8 runs. We also show all individual results as smaller data points.

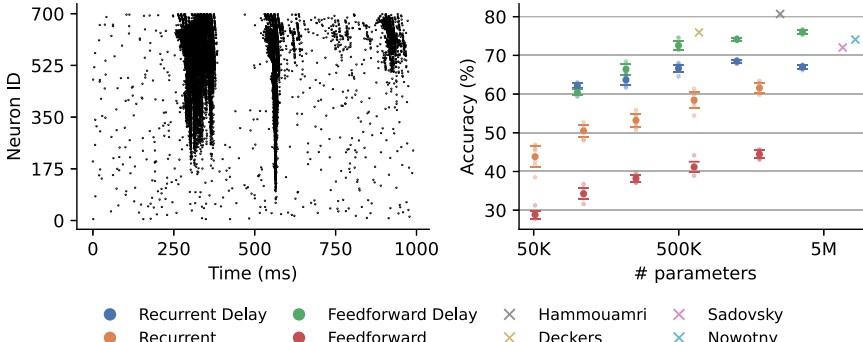

**Fig. 6 | Spiking Speech Commands results.** Left: An example of a speaker saying "nine" from the Spiking Speech Commands (SSC) dataset. Right: For SSC, which has separate training, validation, and test sets, we apply early stopping when validation accuracy no longer improves and report the corresponding test accuracy. Feedforward networks have 2 hidden layers, and recurrent networks have one recurrently connected hidden layer. We implemented models with 128, 256 and 512 hidden neurons. Crosses show results from other delay learning models[27,35,59], and we also show previous Eventprop results by Nowotny, Turner, and Knight[21]. The points show average accuracy, and the error bars show standard deviation over 8 runs. We also show all individual results as smaller data points.

hidden neurons with recurrent connections. The feedforward delays were initialised from a uniform distribution in the range of 0–150 ms, while the recurrent delays were all initialised to 0 ms. The difference between these results and those reported using the 'delay line' approach (93.5 ± 0.7%[21]) are not statistically significant ($p = 0.442$, t-test, $n = 8$), and we achieved them with around 5 times fewer parameters. We also experimented with different hidden layer sizes and feedforward models. We found that decreasing the hidden neuron number to 256 does not significantly decrease the accuracy for either architecture. However, if we decrease the number of neurons even further to 128, we observe a significant drop for the feedforward architecture but not for the recurrent one. Increasing the hidden layer to 1024 neurons leads to overfitting.

While state-of-the-art models – such as the work by Hammouamri, Khalfaoui-Hassani, and Masquelier[27], Deckers et al.[35], and Sun et al.[53] – reported higher accuracies, these were obtained using the test set for validation and early stopping, rather than using a separate validation set. Schöne et al.[54] mention that they also evaluate in this way to achieve a "fair comparison to others". However, as Baronig et al.[55] argued, not only is this not methodologically "clean" but it may also not be entirely fair due to potential overfitting (we note that the highest test accuracy we observed was 95.32%). Furthermore, we also note that model performance on the SHD dataset is nearing saturation, with the best-performing models achieving an accuracy of around 93%. Following Isaksson et al.[56] and Nowotny[57], we calculated the Bayesian confidence intervals with naive assumptions on error rates. 93% accuracy has overlapping confidence intervals with higher accuracies (e.g., 94% and 95%), indicating that further improvements in accuracy are likely not statistically meaningful given the test set size (2264)[57].

## Spiking speech commands

SSC is significantly more challenging than SHD as the audio recordings were created in noisy environments, and the dataset has more classes. We initially experimented with single recurrent hidden layer architectures similar to those employed by Nowotny, Turner, and Knight[21] and, after replacing the delay line inputs with learnable delays, we achieved similar performance. Interestingly, we observed little to no benefit of adding delays to larger networks but, as we decreased the number of hidden neurons, the delays became highly beneficial. While many state-of-the-art models use deeper architectures with more hidden layers[27,55,58], we found that deeper architectures with recurrent connections became highly unstable even without delays in the connections. Therefore, to improve upon previous results, we explored deeper *feedforward* architectures with delays and found the best performing architecture to be a model with 2 feedforward hidden layers.

Our results are shown in Fig. 6. Our best model achieved a training accuracy of 79.6 ± 1.0%, a validation accuracy of 78.1 ± 1.0% ($n = 8$) and a test accuracy of 76.1 ± 1.0%. We also experimented with smaller models, which as expected, achieved a lower training accuracies. Compared to other SNNs with delays, we observe that we outperform Sadovsky, Jakubec, and Jarina[59], with their test accuracy results at 72.03%. Deckers et al.[35] introduced a constrained adaptive LIF neuron model to a delayed network and reached 80.23% test accuracy. They also tested LIF models, achieving 75.94%, which we slightly outperformed.

## Braille letter reading

As highlighted by Walters et al.[60], most neuromorphic benchmarks contain more spatial than temporal information. Although the SHD and SSC datasets *do* contain temporal information, it may not be important enough to require the fine timesteps our approach enables. Therefore, we also evaluated our approach on a braille letter reading dataset[61].

We again trained, validated and tested 2-layer feedforward and single-layer recurrent networks with and without delays and with hidden layers of various sizes (64, 128, 256, 512, 1024) on the 70: 10: 20 training, validation, and test splits provided by Müller-Cleve et al.[61]. The results shown in Fig. 7 show a similar pattern to the SSC results – introducing delays in large recurrent networks shows no benefit, but adding them to smaller networks significantly improves performance.

Our best-performing model had two feedforward hidden layers, with 1024 neurons each, and achieved 83.1 ± 1.5% on the test set. This model outperforms the recurrent network with a single hidden layer of 450 neurons and 8 input copies described in the original publication[61], which obtained a test accuracy of 80.9 ± 0.3%. Our smaller feedforward network with hidden layers of size 256 achieved a test accuracy of 81.0 ± 0.7% so also outperformed Müller-Cleve et al.[61], with half the number of parameters. Additionally, Müller-Cleve et al.[61] evaluated their models using an 80: 20 training-test split without a validation set, which is problematic for the same reasons identified in SHD dataset studies. Pedersen et al.[62] previously trained on this dataset but simplified it to 7 classes to accommodate the Xylo chip[63]. We are not aware of any other research involving delay learning on this dataset.

## Training efficiency

Finally, we benchmarked our training procedure against the Dilated Convolution implementation provided by Hammouamri, Khalfaoui-Hassani, and Masquelier[27] using PyTorch 2.5.1 and SpikingJelly 0.0.0.0.15[64]. Because the Dilated Convolution method does not support recurrent delays, we benchmarked feedforward models with 2 hidden layers. We measured the peak memory utilisation of mlGeNN using the "nvidia-smi" command line tool – as GeNN allocates memory

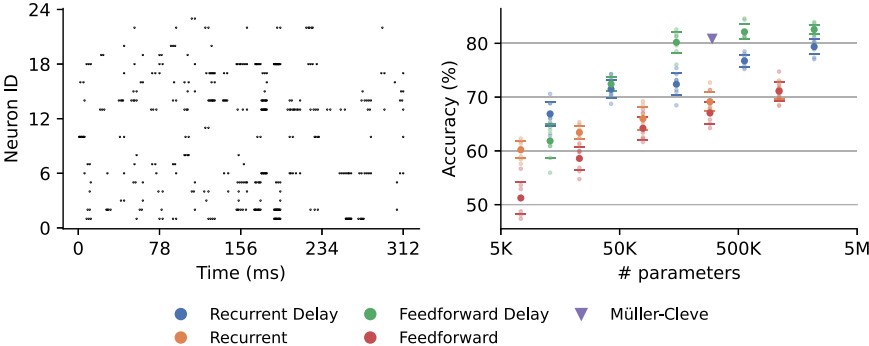

**Fig. 7 | Braille letter reading results.** Left: Example of the letter 'N' from the braille letter reading dataset. We split the dataset into training, validation and test set in a ratio of 70 : 10 : 20, respectively. We tune our hyperparameters based on validation results and report the corresponding test accuracy. Right: Feedforward networks have 2 hidden layers, and recurrent networks have one recurrently connected hidden layer. We implemented models with 128, 256, 512 and 1024 neurons. The triangle depicts validation results reported by Müller-Cleve et al.[61]. The points show average accuracy, and the error bars show standard deviation over 8 runs. We also show results from all individual results with smaller data points.

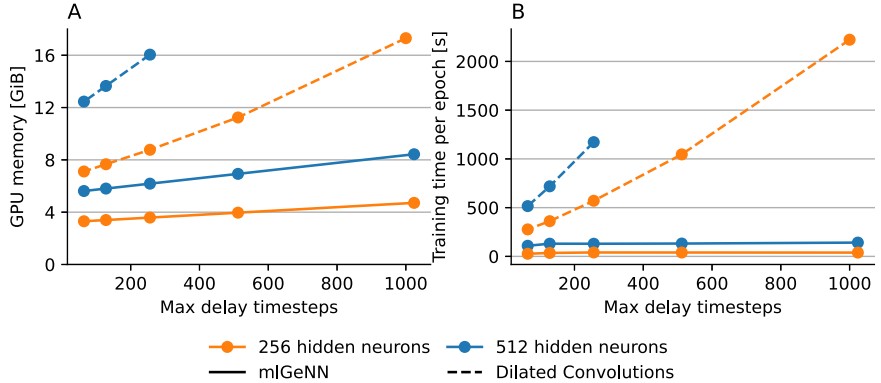

**Fig. 8 | Comparing cost of training networks with delays using EventProp and mlGeNN against Dilated Convolutions (DC) implemented using SpikingJelly and PyTorch. A** Peak GPU memory usage. Missing data points indicate where PyTorch ran out of GPU memory. **B** Time to train one SHD epoch. All experiments were performed on a workstation with an NVIDIA RTX A5000 GPU. All models have two feedforward hidden layers and use batch size 256 and 1 ms timesteps.

from CUDA directly – and of PyTorch using torch.cuda.max_memory_allocated – so details of the memory allocator are disregarded.

The benchmarking results are illustrated in Fig. 8. Because increasing the maximum number of delay slots in mlGeNN only involves increasing the size of a *per-neuron* buffer rather than increasing the size of a *per-synapse* kernel, longer delays have much lower memory requirements in mlGeNN (Fig. 8A). Similarly, the increasing computational cost of convolving larger and larger kernels means that training time increases rapidly when using Dilated Convolutions (Fig. 8B), whereas with mlGeNN, there is only a very small initial increase in training time as the maximum number of delay timesteps increases. Other gradient-based delay learning methods[28,37,38] do not use temporal convolutions, so their memory and computational requirements would not grow *as* quickly, but they are all still based on dense BPTT so – based on benchmarking performed by Nowotny, Turner, and Knight[21] – we would still expect mlGeNN to be significantly faster and use less memory.

The slight increase in mlGeNN training time observed as the maximum number of delay timesteps increases is likely due to the effects of caching on the delay buffers. These buffers are allocated in GPU global memory, so the efficiency of updating them depends on whether they persist in the GPU's caches for long enough for locality in the delays to be exploited. At the large batch sizes used for these experiments, the complete buffers will not fit in the 6 MiB L2 cache of the RTX A5000 GPU so the effects seen here are likely due to an increase in the rate at which delay buffer data gets evicted from the cache as the size of the buffers increases.

## Discussion

Delays in Spiking Neural Networks have been extensively studied from both machine learning and computational neuroscience perspectives, with recent interest spurred by their ability to improve performance in machine learning applications[27,37]. Delays can be efficiently implemented on several neuromorphic chips[33,34], but, as is the case with much of current neuromorphic research, there is a lack of focus on delay *learning* algorithms that could be implemented on future neuromorphic hardware. Instead, many methods rely on arithmetically intensive approaches that are only practical on GPUs, such as convolutions in networks trained with BPTT with surrogate gradients. Our method combines the best of both worlds: its theoretical foundations enable efficient implementation on neuromorphic hardware (EventProp has already been implemented in SpiNNaker 2[33]), while our GeNN-based implementation takes advantage of readily available GPU hardware. This versatility allows us to address a broader range of platforms while improving performance on complex temporal tasks.

The beneficial scaling of EventProp allows long sequence lengths and hence finer timesteps, which might offer enhanced precision in spatio-temporal tasks. However, leading models on SHD and SSC employ large timesteps of 10 or even 25 ms[27,58] and, while these coarser time grids may have been primarily chosen to fit within GPU memory, they could also simplify the tasks by reducing their effective sequence length (assuming high temporal precision is not required). This reflects a broader challenge in SNN research: while neuromorphic architectures are appealing for their energy efficiency and temporal precision, it remains unclear how much temporal precision is needed for

any given task and how it is best exploited in practice. Timestep length also has additional relevance for delay learning, as delays are discretised to integer multiples of the timestep. However, our previous work indicates that the speech recognition tasks considered here may not require precise delays[65].

Another open question regards initialisation. While synaptic weights can be initialised in a straightforward manner by sampling from a normal distribution centred around zero, the initialisation of delays is less intuitive. Experimental observations of delay distributions are challenging to interpret[66], and the optimal distribution may depend heavily on the task. We followed the common approach of initialising delays with values sampled from uniform distributions[27]. However, interestingly, we observed that while a few delays grew considerably, most remained relatively short. This distribution might reflect a small-world network structure, where most neurons connect to nearby neighbours (short delays) with only a few long-range connections (long delays).

As we show across multiple datasets, recurrent delays can offer substantial advantages when implementing networks with stricter size constraints. However, initialising recurrent delay distributions poses additional computational challenges. Initialising feedforward delays within the range $[0, d_{max}]$ is logical, as adding a homogeneous delay $x$ would yield the same outcomes within the range $[x, d_{max} + x]$. However, this symmetry does not extend to recurrent delays, making their initialisation significantly more complex. Recent studies have focused on optimising delay distributions at the network level[67], but the question of layer-specific initialisation remains open.

This work showed the benefit of delays on datasets commonly used to benchmark SNNs. Exploring them in other tasks where they could be particularly beneficial (e.g. sound localisation[68], or motion detection[69,70]) is an interesting avenue for future work.

## Methods
### Theory
**Learning weights in networks with delay**. We start by defining our two differential equations in the implicit form for the membrane potentials and input currents, respectively.

$$\mathbf{f}_V \equiv \tau_m \dot{\mathbf{V}} + \mathbf{V} - \mathbf{I} = 0 \tag{7}$$

$$\mathbf{f}_I \equiv \tau_s \dot{\mathbf{I}} + \mathbf{I} = 0 \tag{8}$$

In the following, we will assume that all event times $\mathscr{E}$ are distinct, both in terms of spikes occurring and of spikes arriving. In continuous time, this is not unlikely, but also, as argued in ref. 18, the equations do not break down if spikes occur or arrive at the same time. Then,

$$\frac{d\mathscr{L}}{dw_{ji}} = \frac{d}{dw_{ji}} \left[ l_p(\mathscr{S}) + \sum_{t_k^{event} \in \mathscr{E}} \int_{t_k^{event}}^{t_{k+1}^{event}} \left[ l_V(\mathbf{V}, t) + \boldsymbol{\lambda}_V \cdot \mathbf{f}_V + \boldsymbol{\lambda}_I \cdot \mathbf{f}_I \right] dt \right] \tag{9}$$

where we have added the product of adjoint variables and dynamics functions to the loss function as the adjoint method dictates. This is possible because for solutions of the forward dynamics, $\mathbf{f}_V$ and $\mathbf{f}_I$ are identically zero at all times. Using

$$\frac{\partial \mathbf{f}_V}{\partial w_{ji}} = \tau_m \frac{d}{dt} \frac{\partial \mathbf{V}}{\partial w_{ji}} + \frac{\partial \mathbf{V}}{\partial w_{ji}} - \frac{\partial \mathbf{I}}{\partial w_{ji}} \tag{10}$$

$$\frac{\partial \mathbf{f}_I}{\partial w_{ji}} = \tau_s \frac{d}{dt} \frac{\partial \mathbf{I}}{\partial w_{ji}} + \frac{\partial \mathbf{I}}{\partial w_{ji}}, \tag{11}$$

we can apply the derivative on the right-hand side of (9) to obtain

$$
\begin{aligned}
\frac{d\mathscr{L}}{dw_{ji}} = & \sum_{t_k^{spike} \in \mathscr{S}} \frac{\partial l_p}{\partial t_k^{spike}} \frac{dt_k^{spike}}{dw_{ji}} \\
& + \sum_{t_k^{event} \in \mathscr{E}} \int_{t_k^{event}}^{t_{k+1}^{event}} \left[ \frac{\partial l_V}{\partial \mathbf{V}} \cdot \frac{\partial \mathbf{V}}{\partial w_{ji}} + \boldsymbol{\lambda}_V \cdot \left( \tau_m \frac{d}{dt} \frac{\partial \mathbf{V}}{\partial w_{ji}} + \frac{\partial \mathbf{V}}{\partial w_{ji}} - \frac{\partial \mathbf{I}}{\partial w_{ji}} \right) \right. \\
& + \left. \boldsymbol{\lambda}_I \cdot \left( \tau_s \frac{d}{dt} \frac{\partial \mathbf{I}}{\partial w_{ji}} + \frac{\partial \mathbf{I}}{\partial w_{ji}} \right) \right] dt \\
& + l_{V,k+1}^- \frac{dt_{k+1}^{event}}{dw_{ji}} - l_{V,k}^+ \frac{dt_k^{event}}{dw_{ji}}
\end{aligned}
\tag{12}
$$

Using partial integration, we can rewrite

$$\int_{t_k^{event}}^{t_{k+1}^{event}} \boldsymbol{\lambda}_V \cdot \frac{d}{dt} \frac{\partial \mathbf{V}}{\partial w_{ji}} dt = - \int_{t_k^{event}}^{t_{k+1}^{event}} \dot{\boldsymbol{\lambda}}_V \cdot \frac{\partial \mathbf{V}}{\partial w_{ji}} dt + \left[ \boldsymbol{\lambda}_V \cdot \frac{\partial \mathbf{V}}{\partial w_{ji}} \right]_{t_k^{event}}^{t_{k+1}^{event}} \tag{13}$$

and

$$\int_{t_k^{event}}^{t_{k+1}^{event}} \boldsymbol{\lambda}_I \cdot \frac{d}{dt} \frac{\partial \mathbf{I}}{\partial w_{ji}} dt = - \int_{t_k^{event}}^{t_{k+1}^{event}} \dot{\boldsymbol{\lambda}}_I \cdot \frac{\partial \mathbf{I}}{\partial w_{ji}} dt + \left[ \boldsymbol{\lambda}_I \cdot \frac{\partial \mathbf{I}}{\partial w_{ji}} \right]_{t_k^{event}}^{t_{k+1}^{event}}. \tag{14}$$

Inserting this into (12), we get

$$
\begin{aligned}
\frac{d\mathscr{L}}{dw_{ji}} = & \sum_{t_k^{spike} \in \mathscr{S}} \frac{\partial l_p}{\partial t_k^{spike}} \frac{dt_k^{spike}}{dw_{ji}} \\
& + \sum_{t_k^{event} \in \mathscr{E}} \int_{t_k^{event}}^{t_{k+1}^{event}} \left[ \left( \frac{\partial l_V}{\partial \mathbf{V}} - \tau_m \dot{\boldsymbol{\lambda}}_V + \boldsymbol{\lambda}_V \right) \cdot \frac{\partial \mathbf{V}}{\partial w_{ji}} + (-\tau_s \dot{\boldsymbol{\lambda}}_I + \boldsymbol{\lambda}_I - \boldsymbol{\lambda}_V) \cdot \frac{\partial \mathbf{I}}{\partial w_{ji}} \right] dt \\
& + \tau_m \left[ \boldsymbol{\lambda}_V \cdot \frac{\partial \mathbf{V}}{\partial w_{ji}} \right]_{t_k^{event}}^{t_{k+1}^{event}} + \tau_s \left[ \boldsymbol{\lambda}_I \cdot \frac{\partial \mathbf{I}}{\partial w_{ji}} \right]_{t_k^{event}}^{t_{k+1}^{event}} \\
& + l_{V,k+1}^- \frac{dt_{k+1}^{event}}{dw_{ji}} - l_{V,k}^+ \frac{dt_k^{event}}{dw_{ji}}
\end{aligned}
\tag{15}
$$

where the last two terms arise from the derivative of the bounds of the integral in the Leibniz rule. We now define the backwards dynamics of the adjoint variables as usual[18],

$$\tau_m \boldsymbol{\lambda}_V' = - \boldsymbol{\lambda}_V - \frac{\partial l_V}{\partial \mathbf{V}} \tag{16}$$

$$\tau_s \boldsymbol{\lambda}_I' = - \boldsymbol{\lambda}_I + \boldsymbol{\lambda}_V \tag{17}$$

which cancels the terms containing $\frac{\partial \mathbf{V}}{\partial w_{ji}}$ and $\frac{\partial \mathbf{I}}{\partial w_{ji}}$, so that we get

$$
\begin{aligned}
\frac{d\mathscr{L}}{dw_{ji}} = & \sum_{t_k^{spike} \in \mathscr{S}} \frac{\partial l_p}{\partial t_k^{spike}} \frac{dt_k^{spike}}{dw_{ji}} + \sum_{t_k^{event} \in \mathscr{E}} \left( l_{V,k}^- \frac{dt_k^{event}}{dw_{ji}} - l_{V,k}^+ \frac{dt_k^{event}}{dw_{ji}} \right. \\
& + \left. \left[ \tau_m \left( \boldsymbol{\lambda}_V^- \cdot \frac{\partial \mathbf{V}^-}{\partial w_{ji}} - \boldsymbol{\lambda}_V^+ \cdot \frac{\partial \mathbf{V}^+}{\partial w_{ji}} \right) + \tau_s \left( \boldsymbol{\lambda}_I^- \cdot \frac{\partial \mathbf{I}^-}{\partial w_{ji}} - \boldsymbol{\lambda}_I^+ \cdot \frac{\partial \mathbf{I}^+}{\partial w_{ji}} \right) \right] \Big|_{t_k^{event}} \right)
\end{aligned}
\tag{18}
$$

The sum over events in $\mathscr{E}$ extends over spike emission times $t_k^{spike}$ and spike arrival times. We first focus on the spike emission times $t_k^{spike}$. Before the jump at $t_k^{spike}$ we have,

$$V_{n(k)}^- - \vartheta = 0, \tag{19}$$

where $n(k)$ denotes the spiking neuron at event $k$. If we take the derivative of this equation, we get, using the chain rule,

$$\frac{\partial V_{n(k)}^-}{\partial w_{ji}} + \dot{V}_{n(k)}^- \frac{\mathrm{d}t_k^{\text{spike}}}{\mathrm{d}w_{ji}} = 0 \tag{20}$$

$$\Rightarrow \quad \frac{\mathrm{d}t_k^{\text{spike}}}{\mathrm{d}w_{ji}} = -\frac{1}{\dot{V}_{n(k)}^-} \frac{\partial V_{n(k)}^-}{\partial w_{ji}}, \tag{21}$$

and after the jump,

$$V_{n(k)}^+ = 0 \tag{22}$$

$$\Rightarrow \quad \frac{\partial V_{n(k)}^+}{\partial w_{ji}} + \dot{V}_{n(k)}^+ \frac{\mathrm{d}t_k^{\text{spike}}}{\mathrm{d}w_{ji}} = 0. \tag{23}$$

Inserting (21) into (23) we obtain as usual[18]

$$\frac{\partial V_{n(k)}^+}{\partial w_{ji}} = \frac{\dot{V}_{n(k)}^+}{\dot{V}_{n(k)}^-} \frac{\partial V_{n(k)}^-}{\partial w_{ji}}. \tag{24}$$

For the current $I_{n(k)}$, there is no jump at $t_k^{\text{spike}}$, and also not in its derivative: $I_{n(k)}^+ = I_{n(k)}^-$ and $\dot{I}_{n(k)}^+ = \dot{I}_{n(k)}^-$ implies

$$\frac{\partial I_{n(k)}^+}{\partial w_{ji}} = \frac{\partial I_{n(k)}^-}{\partial w_{ji}}. \tag{25}$$

Let us now consider what happens at the spike arrival times, when the spike $k$ at $t_k^{\text{spike}}$ is received at all the postsynaptic neurons $m$ at times $t_k^{\text{spike}} + d_{mn(k)}$ (i.e. we look at $\mathcal{E} \setminus \mathcal{S}$). Note that this is where EventProp with delays becomes substantially different from standard EventProp, where spike emission and arrival times are the same. At spike arrival, the input current of the receiving neurons jumps,

$$I_m^+ = I_m^- + w_{mn(k)}. \tag{26}$$

By taking the derivative with respect to $w_{ji}$, we get

$$\frac{\partial I_m^+}{\partial w_{ji}} + \dot{I}_m^+ \frac{\mathrm{d}t_k^{\text{spike}}}{\mathrm{d}w_{ji}} = \frac{\partial I_m^-}{\partial w_{ji}} + \dot{I}_m^- \frac{\mathrm{d}t_k^{\text{spike}}}{\mathrm{d}w_{ji}} + \delta_{in(k)}\delta_{jm}, \tag{27}$$

where we have used that $\frac{\mathrm{d}(t_k^{\text{spike}} + d_{mn(k)})}{\mathrm{d}w_{ji}} = \frac{\mathrm{d}t_k^{\text{spike}}}{\mathrm{d}w_{ji}}$. Now, using the dynamics equations for $\mathbf{I}$, we also have

$$\tau_s \dot{I}_m^+ = \tau_s \dot{I}_m^- - w_{mn(k)}, \tag{28}$$

and hence,

$$\frac{\partial I_m^+}{\partial w_{ji}} = \frac{\partial I_m^-}{\partial w_{ji}} + \tau_s^{-1} w_{mn(k)} \frac{\mathrm{d}t_k^{\text{spike}}}{\mathrm{d}w_{ji}} + \delta_{in(k)}\delta_{jm}$$
$$= \frac{\partial I_m^-}{\partial w_{ji}} + \left[ \frac{1}{\tau_s \dot{V}_{n(k)}^-} w_{mn(k)} \frac{\partial V_{n(k)}^-}{\partial w_{ji}} \right] \Big|_{t_k^{\text{spike}} + d_{mn(k)}} + \delta_{in(k)}\delta_{jm} \tag{29}$$

where we have used (21) to replace $\frac{\mathrm{d}t_k^{\text{spike}}}{\mathrm{d}w_{ji}}$. Since we have $V_m^+ = V_m^-$ for non-spiking neurons,

$$\frac{\partial V_m^+}{\partial w_{ji}} + \dot{V}_m^+ \frac{\mathrm{d}t_k^{\text{spike}}}{\mathrm{d}w_{ji}} = \frac{\partial V_m^-}{\partial w_{ji}} + \dot{V}_m^- \frac{\mathrm{d}t_k^{\text{spike}}}{\mathrm{d}w_{ji}}. \tag{30}$$

From Eq. (26) and the dynamics equations for $\mathbf{V}$ we know

$$\tau_m \dot{V}_m^+ = \tau_m \dot{V}_m^- + w_{mn(k)}. \tag{31}$$

Putting this together, we get

$$\frac{\partial V_m^+}{\partial w_{ji}} = \frac{\partial V_m^-}{\partial w_{ji}} - \tau_m^{-1} w_{mn(k)} \frac{\mathrm{d}t_k^{\text{event}}}{\mathrm{d}w_{ji}} \tag{32}$$

$$= \frac{\partial V_m^-}{\partial w_{ji}} + \left[ \frac{1}{\tau_m \dot{V}_{n(k)}^-} w_{mn(k)} \frac{\partial V_{n(k)}^-}{\partial w_{ji}} \right] \Big|_{t_k^{\text{spike}} + d_{mn(k)}} \tag{33}$$

We now can insert the expressions (21), (25), (24) and (33) into (18) and reorder terms according to which spike the jumps originate from, we get

$$\frac{\mathrm{d}\mathscr{L}}{\mathrm{d}w_{ji}} = \sum_{t_k^{\text{spike}} \in \mathscr{S}} \left[ \frac{\partial V_{n(k)}^-}{\partial w_{ji}} \left[ \tau_m \left( \lambda_{V,n(k)}^- - \frac{\dot{V}_{n(k)}^+}{\dot{V}_{n(k)}^-} \lambda_{V,n(k)}^+ \right) + \frac{1}{\dot{V}_{n(k)}^-} \left( -\frac{\partial l_p}{\partial t_k^{\text{spike}}} + l_V^+ - l_V^- \right) \right] \right.$$
$$\left. + \tau_s(\lambda_{I,n(k)}^- - \lambda_{I,n(k)}^+) \frac{\partial I_{n(k)}^-}{\partial w_{ji}} \right] \Big|_{t_k^{\text{spike}}}$$
$$+ \sum_m \left[ \tau_m(\lambda_{V,m}^- - \lambda_{V,m}^+) \frac{\partial V_m^-}{\partial w_{ji}} + \tau_s(\lambda_{I,m}^- - \lambda_{I,m}^+) \frac{\partial I_m^-}{\partial w_{ji}} \right] \Big|_{t_k^{\text{spike}} + d_{mn(k)}}$$
$$+ \left[ \frac{\partial V_{n(k)}^-}{\partial w_{ji}} \frac{1}{\dot{V}_{n(k)}^-} \right] \Big|_{t_k^{\text{spike}}} \left[ w_{mn(k)}(\lambda_{I,m}^+ - \lambda_{V,m}^+) \right] \Big|_{t_k^{\text{spike}} + d_{mn(k)}} - \left[ \tau_s \delta_{in(k)}\delta_{jm}\lambda_{I,m}^+ \right] \Big|_{t_k^{\text{spike}} + d_{mn(k)}}. \tag{34}$$

Interestingly, after this detailed work, we find that the update of $\lambda_V$ of the spiking neuron is the same as without delays, apart from taking the receiving neurons' corresponding $\lambda_V$ and $\lambda_I$ at the delayed time.

$$\lambda_{V,n(k)}^- = \left[ \frac{\dot{V}_{n(k)}^+}{\dot{V}_{n(k)}^-} \lambda_{V,n(k)}^+ + \frac{1}{\tau_m \dot{V}_{n(k)}^-} \left( \frac{\partial l_p}{\partial t_k^{\text{spike}}} + l_V^- - l_V^+ \right) \right] \Big|_{t_k^{\text{spike}}}$$
$$+ \left[ \frac{1}{\tau_m \dot{V}_{n(k)}^-} \right] \Big|_{t_k^{\text{spike}}} \sum_m w_{mn(k)} \left[ (\lambda_{V,m}^+ - \lambda_{I,m}^+) \right] \Big|_{t_k^{\text{spike}} + d_{mn(k)}} \tag{35}$$

$$\lambda_{V,m}^- = \lambda_{V,m}^+, \text{ if } m \neq n(k) \tag{36}$$

$$\boldsymbol{\lambda}_I^- = \boldsymbol{\lambda}_I^+. \tag{37}$$

The gradient is then given by

$$\frac{\mathrm{d}\mathscr{L}}{\mathrm{d}w_{ji}} = -\tau_s \sum_{t_k^{\text{spike}} \in \mathscr{S}} \delta_{in(k)} \lambda_{I,j} \big|_{t_k^{\text{spike}} + d_{jn(k)}} = -\tau_s \sum_{\{t_k^{\text{spike}} \mid n(k) = i\}} \lambda_{I,j} \big|_{t_k^{\text{spike}} + d_{ji}}. \tag{38}$$

**Learning delays.** In the following, we will derive the gradients for delays $d_{ji}$ similarly to our weight gradient derivations. We start again with the standard approach for the adjoint method,

$$\frac{\mathrm{d}\mathscr{L}}{\mathrm{d}d_{ji}} = \frac{\mathrm{d}}{\mathrm{d}d_{ji}} \left[ l_p(\mathscr{S}) + \sum_{t_k^{\text{event}} \in \mathcal{E}} \int_{t_k^{\text{event}}}^{t_{k+1}^{\text{event}}} \left[ l_V(\mathbf{V}, t) + \boldsymbol{\lambda}_V \cdot \mathbf{f}_V + \boldsymbol{\lambda}_I \cdot \mathbf{f}_I \right] \mathrm{d}t \right] \tag{39}$$

$$\frac{\partial \mathbf{f}_V}{\partial d_{ji}} = \tau_m \frac{\mathrm{d}}{\mathrm{d}t} \frac{\partial \mathbf{V}}{\partial d_{ji}} + \frac{\partial \mathbf{V}}{\partial d_{ji}} - \frac{\partial \mathbf{I}}{\partial d_{ji}} \tag{40}$$

$$\frac{\partial \mathbf{f}_I}{\partial d_{ji}} = \tau_s \frac{\mathrm{d}}{\mathrm{d}t} \frac{\partial \mathbf{I}}{\partial d_{ji}} + \frac{\partial \mathbf{I}}{\partial d_{ji}}. \tag{41}$$

Therefore,

$$
\frac{d\mathcal{L}}{dd_{ji}} = \sum_{t_k^{\text{spike}} \in \mathcal{S}} \frac{\partial l_p}{\partial t_k^{\text{spike}}} \frac{dt_k^{\text{spike}}}{dd_{ji}}
$$
$$
+ \sum_{t_k^{\text{event}} \in \mathcal{E}} \int_{t_k^{\text{event}}}^{t_{k+1}^{\text{event}}} \left[ \frac{\partial l_V}{\partial \mathbf{V}} \cdot \frac{\partial \mathbf{V}}{\partial d_{ji}} + \boldsymbol{\lambda}_V \cdot \left( \tau_{\mathrm{m}} \frac{d}{dt} \frac{\partial \mathbf{V}}{\partial d_{ji}} + \frac{\partial \mathbf{V}}{\partial d_{ji}} - \frac{\partial \mathbf{I}}{\partial d_{ji}} \right) \right.
$$
$$
\left. + \boldsymbol{\lambda}_I \cdot \left( \tau_{\mathrm{s}} \frac{d}{dt} \frac{\partial \mathbf{I}}{\partial d_{ji}} + \frac{\partial \mathbf{I}}{\partial d_{ji}} \right) \right] dt \tag{42}
$$
$$
+ l_{V,k+1}^- \frac{dt_{k+1}^{\text{event}}}{dd_{ji}} - l_{V,k}^+ \frac{dt_k^{\text{event}}}{dd_{ji}}.
$$

Then, using partial integration,

$$
\int_{t_k^{\text{event}}}^{t_{k+1}^{\text{event}}} \boldsymbol{\lambda}_V \cdot \frac{d}{dt} \frac{\partial \mathbf{V}}{\partial d_{ji}} dt = - \int_{t_k^{\text{event}}}^{t_{k+1}^{\text{event}}} \dot{\boldsymbol{\lambda}}_{\mathbf{V}} \cdot \frac{\partial \mathbf{V}}{\partial d_{ji}} dt + \left[ \boldsymbol{\lambda}_V \cdot \frac{\partial \mathbf{V}}{\partial d_{ji}} \right]_{t_k^{\text{event}}}^{t_{k+1}^{\text{event}}} \tag{43}
$$

$$
\int_{t_k^{\text{event}}}^{t_{k+1}^{\text{event}}} \boldsymbol{\lambda}_I \cdot \frac{d}{dt} \frac{\partial \mathbf{I}}{\partial d_{ji}} dt = - \int_{t_k^{\text{event}}}^{t_{k+1}^{\text{event}}} \dot{\boldsymbol{\lambda}}_I \cdot \frac{\partial \mathbf{I}}{\partial d_{ji}} dt + \left[ \boldsymbol{\lambda}_I \cdot \frac{\partial \mathbf{I}}{\partial d_{ji}} \right]_{t_k^{\text{event}}}^{t_{k+1}^{\text{event}}} \tag{44}
$$

and hence,

$$
\frac{d\mathcal{L}}{dd_{ji}} = \sum_{t_k^{\text{spike}} \in S} \frac{\partial l_p}{\partial t_k^{\text{spike}}} \frac{dt_k^{\text{spike}}}{dd_{ji}}
$$
$$
\sum_{t_k^{\text{event}} \in \mathcal{E}} \left[ \int_{t_k^{\text{event}}}^{t_{k+1}^{\text{event}}} \left( \frac{\partial l_V}{\partial \mathbf{V}} - \tau_{\mathrm{m}} \dot{\boldsymbol{\lambda}}_V + \boldsymbol{\lambda}_V \right) \cdot \frac{\partial \mathbf{V}}{\partial d_{ji}} + \left( -\tau_{\mathrm{s}} \dot{\boldsymbol{\lambda}}_I + \boldsymbol{\lambda}_I - \boldsymbol{\lambda}_V \right) \cdot \frac{\partial \mathbf{I}}{\partial d_{ji}} \right] dt
$$
$$
+ \tau_{\mathrm{m}} \left[ \boldsymbol{\lambda}_V \cdot \frac{\partial \mathbf{V}}{\partial d_{ji}} \right]_{t_k^{\text{event}}}^{t_{k+1}^{\text{event}}} + \tau_{\mathrm{s}} \left[ \boldsymbol{\lambda}_I \cdot \frac{\partial \mathbf{I}}{\partial d_{ji}} \right]_{t_k^{\text{event}}}^{t_{k+1}^{\text{event}}} + l_{V,k+1}^- \frac{dt_{k+1}^{\text{event}}}{dd_{ji}} - l_{V,k}^+ \frac{dt_k^{\text{event}}}{dd_{ji}}. \tag{45}
$$

If we now define the adjoint dynamics as usual, the terms in the integral disappear, and we are left with

$$
\frac{d\mathcal{L}}{dd_{ji}} = \sum_{t_k^{\text{spike}} \in \mathcal{S}} \frac{\partial l_p}{\partial t_k^{\text{spike}}} \frac{dt_k^{\text{spike}}}{dd_{ji}}
$$
$$
+ \sum_{t_k^{\text{event}} \in \mathcal{E}} l_{V,k}^- \frac{dt_k^{\text{event}}}{dd_{ji}} - l_{V,k}^+ \frac{dt_k^{\text{event}}}{dd_{ji}} \tag{46}
$$
$$
+ \left[ \tau_{\mathrm{m}} \left( \boldsymbol{\lambda}_V^- \cdot \frac{\partial \mathbf{V}^-}{\partial d_{ji}} - \boldsymbol{\lambda}_V^+ \cdot \frac{\partial \mathbf{V}^+}{\partial d_{ji}} \right) + \tau_{\mathrm{s}} \left( \boldsymbol{\lambda}_I^- \cdot \frac{\partial \mathbf{I}^-}{\partial d_{ji}} - \boldsymbol{\lambda}_I^+ \cdot \frac{\partial \mathbf{I}^+}{\partial d_{ji}} \right) \right] \Big|_{t_k^{\text{event}}}.
$$

Let's now again first consider the spike emission times $t_k^{\text{spike}}$ and the spiking neuron $n(k)$. Before the jump:

$$
\frac{\partial V_{n(k)}^-}{\partial d_{ji}} + \dot{V}_{n(k)}^- \frac{dt_k^{\text{spike}}}{dd_{ji}} = 0 \tag{47}
$$

$$
\Rightarrow \quad \frac{dt_k^{\text{spike}}}{dd_{ji}} = - \frac{1}{\dot{V}_{n(k)}^-} \frac{\partial V_{n(k)}^-}{\partial d_{ji}}, \tag{48}
$$

and after the jump:

$$
\frac{\partial V_{n(k)}^+}{\partial d_{ji}} + \dot{V}_{n(k)}^+ \frac{dt_k^{\text{spike}}}{dd_{ji}} = 0 \tag{49}
$$

$$
\Rightarrow \quad \frac{\partial V_{n(k)}^+}{\partial d_{ji}} = \frac{\dot{V}_{n(k)}^+}{\dot{V}_{n(k)}^-} \frac{\partial V_{n(k)}^-}{\partial d_{ji}}. \tag{50}
$$

There is no jump in $I_{n(k)}$ or its time derivative at $t_k^{\text{spike}}$ which analogous to above implies

$$
\frac{\partial I_{n(k)}^+}{\partial d_{ji}} = \frac{\partial I_{n(k)}^-}{\partial d_{ji}}. \tag{51}
$$

Turning to spike arrival times $t_k^{\text{event}} \in \mathcal{E} \backslash \mathcal{S}$, when the spike at $t_k^{\text{spike}}$ arrives at the post-synaptic neurons $m$, we get

$$
I_m^+ = I_m^- + w_{mn(k)}, \tag{52}
$$

and hence,

$$
\frac{\partial I_m^+}{\partial d_{ji}} + \dot{I}_m^+ \frac{dt_k^{\text{event}}}{dd_{ji}} = \frac{\partial I_m^-}{\partial d_{ji}} + \dot{I}_m^- \frac{dt_k^{\text{event}}}{dd_{ji}}. \tag{53}
$$

Using the dynamics of $\mathbf{I}$, (52) implies

$$
\tau_{\mathrm{s}} \dot{I}_m^+ = \tau_{\mathrm{s}} \dot{I}_m^- - w_{mn(k)}, \tag{54}
$$

and hence

$$
\frac{\partial I_m^+}{\partial d_{ji}} = \frac{\partial I_m^-}{\partial d_{ji}} + \tau_{\mathrm{s}}^{-1} w_{mn(k)} \frac{dt_k^{\text{event}}}{dd_{ji}} \tag{55}
$$

$$
= \frac{\partial I_m^-}{\partial d_{ji}} - \frac{1}{\tau_{\mathrm{s}} \dot{V}_{n(k)}^-} w_{mn(k)} \frac{\partial V_{n(k)}^-}{\partial d_{ji}} + \delta_{in(k)} \delta_{jm} \frac{w_{mn(k)}}{\tau_{\mathrm{s}}}, \tag{56}
$$

where the term involving the spiking neuron $n(k)$ stems from the derivative of the spike time $t_k^{\text{event}}$ with respect to $d_{ji}$ using (48) and the last term from the derivative of the delay by itself (since $\frac{\partial t_k^{\text{event}}}{\partial d_{ji}} = \frac{\partial (t_k^{\text{spike}} + d_{ji})}{\partial d_{ji}} = \frac{\partial t_k^{\text{spike}}}{\partial d_{ji}} + 1$). Note that this is where the derivations begin to differ from when we were taking the derivative with respect to $w_{ji}$. For the voltages,

$$
\frac{\partial V_m^+}{\partial d_{ji}} + \dot{V}_m^+ \frac{dt_k^{\text{event}}}{dd_{ji}} = \frac{\partial V_m^-}{\partial d_{ji}} + \dot{V}_m^- \frac{dt_k^{\text{event}}}{dd_{ji}}, \tag{57}
$$

and using the dynamics of $\mathbf{V}$ and (52),

$$
\tau_{\mathrm{m}} \dot{V}_m^+ = \tau_{\mathrm{m}} \dot{V}_m^- + w_{mn(k)}, \tag{58}
$$

which put together gives

$$
\frac{\partial V_m^+}{\partial d_{ji}} = \frac{\partial V_m^-}{\partial d_{ji}} - \tau_{\mathrm{m}}^{-1} w_{mn} \frac{dt_k^{\text{event}}}{dd_{ji}} \tag{59}
$$

$$
= \frac{\partial V_m^-}{\partial d_{ji}} + \frac{1}{\tau_{\mathrm{m}} \dot{V}_{n(k)}^-} w_{mn(k)} \frac{\partial V_{n(k)}^-}{\partial d_{ji}} - \delta_{in(k)} \delta_{jm} \frac{w_{mn(k)}}{\tau_{\mathrm{m}}}, \tag{60}
$$

where the last term again arises from the derivative of the delay $d_{mn(k)}$ in $t_k^{\text{event}}$ with respect to $d_{ji}$. Taking everything together, we get

$$
\frac{d\mathcal{L}}{dd_{ji}} = \sum_{t_k^{\text{spike}} \in \mathcal{S}} \left[ \frac{\partial V_{n(k)}^-}{\partial d_{ji}} \left[ \tau_{\mathrm{m}} \left( \lambda_{V,n(k)}^- - \frac{\dot{V}_{n(k)}^+}{\dot{V}_{n(k)}^-} \lambda_{V,n(k)}^+ \right) + \frac{1}{\dot{V}_{n(k)}^-} \left( -\frac{\partial l_p}{\partial t_k^{\text{spike}}} + l_V^+ - l_V^- \right) \right] \right. \tag{61}
$$

$$
+ \tau_{\mathrm{s}} (\lambda_{I,n(k)}^- - \lambda_{I,n(k)}^+) \frac{\partial I_n^-}{\partial d_{ji}} \Big|_{t_k^{\text{spike}}} \tag{62}
$$

$$
\left. + \sum_m \left[ \tau_{\mathrm{m}} (\lambda_{V,m}^- - \lambda_{V,m}^+) \frac{\partial V_m^-}{\partial d_{ji}} + \tau_{\mathrm{s}} (\lambda_{I,m}^- - \lambda_{I,m}^+) \frac{\partial I_m^-}{\partial d_{ji}} \right] \Big|_{t_k^{\text{spike}} + d_{mn(k)}} \right] \tag{63}
$$

**Table 2 | Yin-Yang parameters**

| Architecture | Feedforward no delays | Feedforward with delays |
|---|---|---|
| Number of hidden layers | 1 | 1 |
| Number of hidden neurons | 30/25/20/15/10/5 | 30/25/20/15/10/5 |
| Input-to-hidden weight init. | $\mathcal{N}(1.0, 1.0)$ | $\mathcal{N}(1.0, 1.0)$ |
| Hidden-to-out weight init. | $\mathcal{N}(1.0, 1.0)$ | $\mathcal{N}(1.0, 1.0)$ |
| Ff delay init. | ✗ | $\mathcal{U}(0, 0)$ |
| DT | 0.01/0.005/0.005/0.005/0.005/0.01 | 0.01/0.005/0.005/0.005/0.005/0.01 |
| Weight LR | 0.005/0.01/0.015/0.01/0.02/0.02 | 0.02/0.02/0.015/0.015/0.01/0.015 |
| Weight LR schedule | 0.9975 × epoch | 0.9975 × epoch |
| Delay LR init. | ✗ | 10/6/6/10/5/10 |
| Delay LR schedule | ✗ | 0.9975 × epoch |

$$+ \left[\frac{\partial V_n^-}{\partial d_{ji}} \frac{1}{\dot{V}_{n(k)}^-}\right]\bigg|_{t_k^{\text{spike}}} \left[w_{mn(k)}(\lambda_{I,m}^+ - \lambda_{V,m}^+)\right]\big|_{t_k^{\text{spike}} + d_{mn(k)}} \quad (64)$$

$$- \left[w_{mn(k)}\delta_{in(k)}\delta_{jm}(\lambda_{I,m}^+ - \lambda_{V,m}^+)\right]\big|_{t_k^{\text{spike}} + d_{mn(k)}}. \quad (65)$$

So, using the usual trick

$$\frac{\dot{V}_{n(k)}^+}{\dot{V}_{n(k)}^-} = \frac{\vartheta}{\tau_m \dot{V}_{n(k)}^-} + 1, \quad (66)$$

we again arrive at the same jump conditions as usual,

$$\lambda_{V,n(k)}^- = \left[\lambda_{V,n(k)}^+ + \frac{1}{\tau_m \dot{V}_{n(k)}^-}\left[\vartheta \cdot \lambda_{V,n(k)}^+ + \frac{\partial l_p}{\partial t_k^{\text{spike}}} + l_V^- - l_V^+\right]\right]\bigg|_{t_k^{\text{spike}}}$$
$$+ \left[\frac{1}{\tau_m \dot{V}_{n(k)}^-}\right]\bigg|_{t_k^{\text{spike}}} \sum_m w_{mn(k)}\left[\lambda_{V,m}^+ - \lambda_{I,m}^+\right]\big|_{t_k^{\text{spike}} + d_{mn(k)}} \quad (67)$$

$$\lambda_{V,m}^- = \lambda_{V,m}^+, \text{ if } m \neq n(k) \quad (68)$$

$$\boldsymbol{\lambda}_I^- = \boldsymbol{\lambda}_I^+, \quad (69)$$

but the gradient updates take the form

$$\frac{d\mathcal{L}}{dd_{ji}} = -\sum_{t_k^{\text{spike}} \in \mathcal{S}} w_{ji}\delta_{in(k)}(\lambda_{I,j} - \lambda_{V,j})|_{t_k^{\text{spike}} + d_{jn(k)}}$$
$$= -w_{ji}\sum_{\{t_k^{\text{spike}} | n(k) = i\}} (\lambda_{I,j} - \lambda_{V,j})|_{t_k^{\text{spike}} + d_{ji}}. \quad (70)$$

**Time-invariant mean squared error loss.** Following Göltz et al.[39], we use the time-invariant mean squared error loss of output spike times for the Yin-Yang benchmark

$$\mathcal{L}_{\Delta\text{MSE}} = \frac{1}{2}\sum_{i \neq c}(t_i - t_c - \Delta_t)^2, \quad (71)$$

where $c$ is the true class of the current input and $t_i$, $t_c$ denote the first spike time in the respective output neurons. In the EventProp formalism, this is a spike-time dependent loss $l_p$ and, therefore, drives jumps in $\boldsymbol{\lambda}_{V,i}$ in output neuron $i$ at spike times $t_k^{\text{spike}}$ in the backward pass (see Table 1) by

$$\frac{\partial l_p}{\partial t_k^{\text{spike}}} = \begin{cases} (t_i - t_c - \Delta_t) & \text{if } n(k) = i, t_k^{\text{spike}} = t_i, i \neq c \\ \sum_{i \neq c} -(t_i - t_c - \Delta_t) & \text{if } n(k) = c, t_k^{\text{spike}} = t_c \\ 0 & \text{otherwise} \end{cases} \quad (72)$$

**Table 3 | SHD parameters**

| Architecture | Recurrent | Feedforward |
|---|---|---|
| Number of hidden layers | 1 | 2 |
| Number of hidden neurons | 1024/512/256/128 | 1024/512/256/128 |
| Spike reg. strength (layer-wise) | $5 \cdot 10^{-11}$ | $5 \cdot 10^{-12}$, $5 \cdot 10^{-11}$ |
| Input-to-hidden weight init. | $\mathcal{N}(0.03, 0.01)$ | $\mathcal{N}(0.03, 0.01)$ |
| Ff hidden-to-hidden weight init. | ✗ | $\mathcal{N}(0.02, 0.03)$ |
| Rec hidden-to-hidden weight init. | $\mathcal{N}(0.0, 0.02)$ | ✗ |
| Hidden-to-out weight init. | $\mathcal{N}(0.02, 0.03)$ | $\mathcal{N}(0.02, 0.03)$ |
| Ff delay init. | $\mathcal{U}(0, 150)$ | $\mathcal{U}(0, 100)$ |
| Rec delay init. | $\mathcal{U}(0, 0)$ | ✗ |
| DT | 1.0 | 1.0 |
| Weight LR | 0.001 | 0.001 |
| Weight LR schedule | $1.05^{batch}$ | $1.05^{batch}$ |
| Delay LR init. | 0.1 | 0.1 |
| Delay LR schedule | ✗ | ✗ |

## Implementation

We implemented all of our work in the mlGeNN framework[40,41] to exploit the the efficiency of event-based learning. In all of our experiments, we used the parameters from previous EventProp work[21], apart from spike regularisation strengths, number of hidden layers and recurrent connections. We did not implement heterogeneous and trainable time constants, so that the independent effect of delays would be more clear. For our experiments on the SHD and SSC datasets, we adopted the data augmentation approaches described by Nowotny, Turner, and Knight[21], which were designed to improve generalization. Specifically, we implemented the following augmentations:

- Input Shifting: We randomly shifted all inputs by a value within the range of (−40, 40).
- Input Blending: We blended two inputs from the same class by aligning their centres of mass and randomly selecting spikes from each input with a probability of 0.5.

For SSC we only used the shift augmentation. For the Yin-Yang dataset we decreased the learning rate on both weights and delays at the end of each epoch. On SHD and SSC, we implemented an "ease-in" scheduler on the weight learning rate, starting from 0.001 times the learning rate, increasing it at the end of each batch, until it reached the final value. For our chosen hyperparameters, see Tables 2–5. GeNN already provided an efficient implementation of spike transmission with per-synapse delays[43] – allowing the EventProp forward pass to be implemented efficiently. However, the $\boldsymbol{\lambda}_V$ transitions in the backward pass require access to postsynaptic $\boldsymbol{\lambda}$ values with a per-synapse delay ($[\lambda_{V,m}^+ - \lambda_{I,m}^+]|_{t_k^{\text{spike}} + d_{mn(k)}}$ from Equation: (35)). This required a small

**Table 4 | SSC parameters**

| Architecture | Recurrent | Feedforward |
|---|---|---|
| Number of hidden layers | 1 | 2 |
| Number of hidden neurons | 1024/512/256/128/64 | 1024/512/256/128/64 |
| Spike reg. strength (layer-wise) | $5 \cdot 10^{-11}$ | $5 \cdot 10^{-12}, 5 \cdot 10^{-12}$ |
| Input-to-hidden weight init. | $\mathcal{N}(0.03, 0.01)$ | $\mathcal{N}(0.03, 0.01)$ |
| Ff hidden-to-hidden weight init. | ✗ | $\mathcal{N}(0.02, 0.03)$ |
| Rec hidden-to-hidden weight init. | $\mathcal{N}(0.0, 0.02)$ | ✗ |
| Hidden-to-out weight init. | $\mathcal{N}(0.02, 0.3)$ | $\mathcal{N}(0.02, 0.03)$ |
| Ff delay init. | $\mathcal{U}(0, 150)$ | $\mathcal{U}(0, 50)$ |
| Rec delay init. | $\mathcal{U}(0, 0)$ | ✗ |
| DT | 1.0 | 1.0 |
| Weight LR | 0.001 | 0.001 |
| Weight LR schedule | $1.05^{batch}$ | $1.05^{batch}$ |
| Delay LR init. | 0.1 | 0.1 |
| Delay LR schedule | ✗ | ✗ |

**Table 5 | Braille reading parameters**

| Architecture | Recurrent | Feedforward |
|---|---|---|
| Number of hidden layers | 1 | 2 |
| Number of hidden neurons | 1024/512/256/128/64 | 1024/512/256/128/64 |
| Spike reg. strength (layer-wise) | $10^{-10}/^{-10}/^{-8}/^{-8}/^{-8}$ | $10^{-10}, 10^{-10}$ |
| Input-to-hidden weight init. | $\mathcal{N}(0.03, 0.01)$ | $\mathcal{N}(0.03, 0.01)$ |
| Ff hidden-to-hidden weight init. | ✗ | $\mathcal{N}(0.02, 0.03)$ |
| Rec hidden-to-hidden weight init. | $\mathcal{N}(0.0, 0.02)$ | ✗ |
| Hidden-to-out weight init. | $\mathcal{N}(0.02, 0.03)$ | $\mathcal{N}(0.02, 0.03)$ |
| Ff delay init. | $\mathcal{U}(0, 0)$ | $\mathcal{U}(0, 0)$ |
| Rec delay init. | $\mathcal{U}(0, 0)$ | ✗ |
| DT | 4.0 | 4.0 |
| Weight LR | 0.0015 | 0.0015 |
| Weight LR schedule | ✗ | ✗ |
| Delay LR init. | 0.025 | 0.025 |
| Delay LR schedule | ✗ | ✗ |

extension to GeNN's existing system for providing delayed access to postsynaptic variables from a synapse model[42] in order to enable it to use the per-synapse delays used for spike transmission in the forward pass.

## Data availability
The data underlying our results are available at https://doi.org/10.25377/sussex.29414015.

## Code availability
All experiments were carried out using the GeNN 5.1.0[71] an mlGeNN 2.3.0[72]. The latest versions of both libraries are also available at https://github.com/genn-team/. The code to train and evaluate the models described in this work are available at https://doi.org/10.5281/zenodo.17236061.

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

## Acknowledgements

This work was funded by the EPSRC (grants EP/V052241/1 and EP/S030964/1) and the EU (grant no. 945539). Additionally, B.M. was funded by a Leverhulme Trust studentship and by The Alan Turing Institute. Compute time was provided through Gauss Centre for Supercomputing (application numbers 21018, 30182 and 61883) and EPSRC (grant number EP/T022205/1) and local GPU hardware was provided by an NVIDIA hardware grant award.

## Author contributions

B.M. and T.N. jointly developed the theory. J.C.K. implemented the resulting new methods into GeNN and mlGeNN. B.M. designed, implemented and executed the numerical experiments. J.C.K. implemented and executed the training efficiency tests. J.C.K. and T.N. supervised the project. B.M., J.C.K and T.N. wrote and revised the manuscript.

## Competing interests

The authors declare no competing interests.
