## [Transparent Peer Review file · Nature Communications]

Efficient Event-based Delay Learning in Spiking Neural Networks

Corresponding Author: Mr Balázs Mészáros

Version 0:

Reviewer comments:

Reviewer #2

(Remarks to the Author)

In their paper Efficient Event-based Delay Learning in Spiking Neural Networks, Mészáros et al. describe an extension to the EventProp formalism, augmenting the latter by trainable synaptic delays.

The authors, further, benchmark SNNs with and without delays to demonstrate the advantage of these additional degrees of freedom in terms of classification accuracy.

They also compare their approach -- in general only relying on sparse information for the backward pass -- to the work of Hammouamri et al., highlighting the increase in computational efficiency when relying on sparse gradients.

The authors present an incremental, but yet significant advance in the field of SNN training schemes, and I would be generally inclined to recommend publication of the manuscript (assuming the issues below are addressed).

The paper is concisely written and adequately introduces the concept spiking neural networks and the potential power of synaptic transmission delays as additional degrees of freedom.

The benchmarking seems mostly sound and I have no significant methodological complaints.

Unfortunately, I have neither the background nor the capacity to review the mathematical methods.

I fully trust the authors to have correctly derived the delay-aware training adjoint dynamics and their results are backed up and in line with other delay learning approaches, but I can not exclude any issues hidden in the equations.

The authors should thus double-check their math.

The manuscript could profit from an extended discussion of event-based and dense computation and in that context also differentiate between event-based frameworks -- i.e., a mathematical description only relying on computation at the time of (delayed) spike events -- and an event-based implementation.

This is of particular importance, as the presented implementation is (very likely deliberately) not following that paradigm.

Especially considering the interdisciplinary audience of the journal, I would also like to see a more gentle introduction into the EventProp formalism.

The authors already try to hide a lot of the complexity by moving the derivation into the methods section, but confronting the reader with non-trivial math already at the beginning of the results section as well as in Table 1 seems a bit brutal.

Considering the fact that the extended formalism is the primary result of the manuscript, I would not recommend to completely hide away the math, but the authors should think of a more friendly way of introducing it.

I would, in that context, also recommend to more clearly highlight their own extensions to the delay-less adjoint dynamics, especially in Table 1.

Contrasting the impact of delays with the already published state-of-the-art -- and assuming that the resulting modifications are straight-forward to highlight -- might make that actual delta easier to digest.

The same also applies to Figure 1, which is similarly overwhelming.

It might be advisable to add a new Figure 1 that gently introduces the main concepts without going into the details.

I enjoyed the fact that the authors benchmarked their formalism for different network topologies and on multiple datasets. However, there are a few minor issues I would like to see addressed:

1. The Yin-Yang dataset, in its original form, only features four input dimensions and does not include a bias signal. The latter is, unfortunately, often introduced as an additional timing signal and might even be required for some forms of loss functions.

Göltz et al. don't seem to be relying on a bias signal for their benchmarking.

Considering the fact that DelGrad is -- under certain constraints -- very similar in nature to the present work, I would like to see a more direct comparison of the two.

For delay-less networks of LIF neurons, EventProp and the formalism behind DelGrad, seem to result in identical gradients. Including that direct comparison, i.e., training mostly identical networks on identical data, would be potentially interesting to the reader.

2. I don't fully understand the reasoning behind consistently showing training, validation, and testing accuracy.

It might lead to interesting observations w.r.t. potential overfitting, but also introduces noise.

3. The authors discuss the fact that other studies rely on SHD's test set for model selection and final performance evaluation. This is generally a very relevant issue, and Hammouamri et al. even clearly state doing so in their manuscript.

I am not sure, however, about the work by Schöne et al.

The authors should clearly point to the methodological issues and trace their claims back to the respective sources ("as XY et al. indicate in their manuscript, they...").

4. I would recommend the authors to use a LaTeX package like `\siunitx` to correctly typeset all unit-carrying quantities.

This also prevents a mismatch of significant digits, e.g., in accuracy figures ("98.47±0.004%").

5. In their performance benchmarks, the authors mention the "likely" impact of caching.

I would like to see a less vague explanation, especially considering the fact that the authors also develop the underlying software framework.

6. I would like to see the network performance plotted against the number of network parameters.

This likely requires a more extensive sweep across network sizes than what is currently shown in Figures 3, 4, and 5.

Generally, I find the extensive use of bar plots suboptimal, mostly due to their low information density.

Line plots with one line for weight-only and another for delay-augmented networks improve legibility and comparability.

Along those lines, I would also like to see weight-only data for the SHD and SSC datasets.

Apart from those points, I would also like the authors to address a number of mostly cosmetic and stylistic issues:

1. Ensure correct and homogeneous typesetting on non-counted/non-variable subscripts.

2. Mathematical operators, i.e., the differential operator $d\Box$, are to be set upright.

I'd recommend using a suitable LaTeX package for that.

3. The figures could profit from a bit of love.

This is especially true for Figure 3 B, where the spikes are not clearly visible, and the respective labels seem to be misplaced.

The figure could be generally improved by connecting panels A and B, see, e.g., the original publication on the Yin-Yang dataset.

4. The abstract starts with the theoretical but somewhat vague claim that SNNs are more efficient than ANNs, and inherently better suited than traditional AI methods for time series processing.

5. The introduction starts similarly generic and vague.

Motivating delay learning on keyword spotting tasks with the energy consumption of training LLMs seems somewhat of a stretch.

This vague and somewhat "populist" introduction does not do the scientific work addressed in this manuscript justice.

6. Gradient-based training of SNNs is introduced with the issue of the "non-differentiable Heaviside activation function".

Thresholding in a time-continuous system does not imply the use of a Heaviside function, which is merely a misunderstood implementation detail of time-stepped numerical integration of SNNs.

And the adjoint formalism correctly addresses the underlying issue, as the authors surely know and understand.

This paragraph should be reworked to avoid this common misconception.

7. The manuscript occasionally relies on colloquial language (e.g., "... into something differentiable.").

8. The authors should more concise when talking about the "exploding memory requirements" and the resulting limits to sequence lengths when using surrogate gradients.

9. In section 2.4, the authors mention "spike regularizations strength" without ever introducing the concept.

References

Göltz, Julian, Jimmy Weber, Laura Kriener, Peter Lake, Melika Payvand, and Mihai A Petrovici. 2024. "Delgrad: Exact Gradients in Spiking Networks for Learning Transmission Delays and Weights". Arxiv E-Prints, arXiv-2404.

Hammouamri, Ilyass, Ismail Khalfaoui-Hassani, and Timothée Masquelier. 2023. "Learning Delays in Spiking Neural Networks Using Dilated Convolutions with Learnable Spacings". Arxiv Preprint Arxiv:2306.17670.

Schöne, Mark, Neeraj Mohan Sushma, Jingyue Zhuge, Christian Mayr, Anand Subramoney, and David Kappel. 2024. "Scalable Event-by-Event Processing of Neuromorphic Sensory Signals with Deep State-Space Models". In 2024 International Conference on Neuromorphic Systems (ICONS), 124–31.

(Remarks on code availability)

(Remarks to the Author)

The authors extend EventProp, an algorithm for event-based training of Spiking Neural Networks (SNNs) to include the training of synaptic delays. They compare their method to the state of the art for SNNs on two data sets and compare the memory usage and training time to an alternative method, that is, dilated convolutions.

The proposed method is interesting and to the best of my knowledge novel. However, it is a relatively straight-forward extension of EventProp, so it is somewhat incremental. In addition to that, the evaluation of the method is quite rudimentary and does not demonstrate a clear advantage of the model over existing ones. My major points are:

- 1) The method is benchmarked only on two standard data sets: SHD and Spiking Speech Commands. On the former, it is very close to or on par with SOTA models. As the authors correctly mention, the performance on this data set has already quite saturated. On the latter, it is clearly below the SOTA for SNNs. Therefore, it is not clear where the advantage of the approach lies. There are several other data sets that could be used for a comparison. The method is not compared in terms of accuracy to other methods for delay learning. Besides the elegance, no measurable advantages are reported.
- 2) There are many papers on delay learning in SNNs. The authors discuss only a few of them. For example see works cited in their ref [27] and later ones. Performance is only compared to one model in terms of memory usage and training time, where it is relatively clear that the method should have an advantage. But how does it compare to other methods for delay learning such as Slayer or others?
- 3) The manuscript is very hard to understand if one does not have background knowledge about the adjoint method. The authors could make an attempt to describe their method (and derivation) with more provided intuition and background. Writing could be improved:
 - I_p and I_V are not well described. I could not find a definition of the actual functions for the simulations.
 - It is not described what t_k stands for - I assume $\{t_k\}$ stands for the set of all spikes in the network. Maybe it would be better to adopt a different notation for that. The set notation looks a bit sloppy to me, in particular in eq. (4) where the delay is also incorporated. It is not well defined how this notation works.
 - In Fig. 1 there is a t_1^{\max} which is not defined. Also t^{post} in eq. (10) is not defined.
- 4) The figures are below usual standards and hard to parse. For a high class journal, I believe that also nice figures are needed.
 - Fig 1: It is very hard to understand from the figure and the description what is happening here. Notation is not well-described (such as d^{hi}_{10} etc)
 - Fig 3: is not acceptable in my opinion (in particular 3B and 3C). Also, the encoding shown in Fig. 3B is not described.
 - Axes labels missing in Fig. 4, 5.

Minor points:

- In 2.2: It seems that the authors are not using their algorithm but rather DelGrad?
- There are some smaller typos (e.g. τ_{mem} and τ_{syn} in the legend of Table 1).
- lines 44, 45: Surrogate gradients do not 'smooth out' spikes. They assume a smoothed threshold function for the gradient.
- Punctuation around equations is often not correct.

(Remarks on code availability)

Version 1:

Reviewer comments:

Reviewer #2

(Remarks to the Author)

The authors have mostly addressed my previous concerns, I appreciate it.

I am still unsure about Figs. 1 & 2. It might be advisable to invest some work into creating an overview figure (as Fig. 1) to summarize the storyline of the manuscript. This might be more of an editorial decision.

I have one main concern, and that is the performance reported on the Yin-Yang dataset.

Göltz et al. report that delays improve performance at iso-parameter count, and in general, the manuscript seems to report significantly lower performance when compared to DelGrad.

The authors should either discuss potential reasons for this discrepancy or ensure identical set ups.

Furthermore, I would recommend to avoid the use of abbreviations, especially in the legends of Figs. 4 to 7.

Lastly, section 2.6 could profit from an improved title. "Computational performance" can refer to both, the computational capabilities of the algorithm and also its cost.

(Remarks on code availability)

Reviewer #3

(Remarks to the Author)

The authors have provided an improved revised manuscript. Figures and mathematical description have improved.

Pros:

- The proposed method is novel and interesting.
- Its efficiency in terms of training time per epoch is clearly better than a previous method based on dilated convolutions, although other methods were not compared.

Cons:

- It is a relatively straight-forward extension of EventProp, so it is somewhat incremental.
- The performance in terms of accuracy is close to the state of the art for SNNs but does not improve beyond that.
- The authors have improved the mathematical description, but could still be improved. For example
- It is not mentioned what t_k^{spike} denotes exactly (is $t_1^{\text{spike}}, t_2^{\text{spike}}, \dots$ the (sorted?) list of all spikes in the network?)
- I did not find a definition of $e_n(k)$ in Table 1.
- The notation " t_k^{spike} from i " is sloppy.
- Are V, I, \dots vectors? This is never explicitly stated.
- The mathematical description in Methods is very technical. The authors did not make an attempt to describe their derivation with more provided intuition and background there.

In summary, it is a solid paper with some weak points, and there is still some potential for improvement regarding the writing.

(Remarks on code availability)

Version 2:

Reviewer comments:

Reviewer #3

(Remarks to the Author)

The authors have submitted a revised version with some small changes, addressing my suggestions.

Overall, the manuscript did not change much, and my assessment is unchanged: The paper is solid with some pros and cons for publication in Nature Communications.

Pros:

- The proposed method is novel and interesting.
- Its efficiency in terms of training time per epoch is clearly better than a previous method based on dilated convolutions, although other methods were not compared.

Cons:

- It is a relatively straight-forward extension of EventProp, so it is somewhat incremental.
- The performance in terms of accuracy is close to the state of the art for SNNs but does not improve beyond that.
- The mathematical description in Methods is very technical, although I understand that it is difficult to balance the didactic narrative with compactness.

(Remarks on code availability)

Reviewer #2 (Remarks to the Author):

The manuscript could profit from an extended discussion of event-based and dense computation and in that context also differentiate between event-based frameworks -- i.e., a mathematical description only relying on computation at the time of (delayed) spike events -- and an event-based implementation.

This is of particular importance, as the presented implementation is (very likely deliberately) not following that paradigm.

We now discuss this when we introduce mIGeNN at lines 88-105.

Especially considering the interdisciplinary audience of the journal, I would also like to see a more gentle introduction into the EventProp formalism.

The authors already try to hide a lot of the complexity by moving the derivation into the methods section, but confronting the reader with non-trivial math already at the beginning of the results section as well as in Table 1 seems a bit brutal.

We have added an additional, less formal, introduction to the EventProp algorithm at lines 120-138.

Considering the fact that the extended formalism is the primary result of the manuscript, I would not recommend to completely hide away the math, but the authors should think of a more friendly way of introducing it.

I would, in that context, also recommend to more clearly highlight their own extensions to the delay-less adjoint dynamics, especially in Table 1.

Contrasting the impact of delays with the already published state-of-the-art -- and assuming that the resulting modifications are straight-forward to highlight -- might make that actual delta easier to digest.

We added the full equations to Table 1 and coloured the differences to the existing algorithm in red to highlight the changes.

The same also applies to Figure 1, which is similarly overwhelming.

It might be advisable to add a new Figure 1 that gently introduces the main concepts without going into the details

Apologies for this somewhat overwhelming start to the paper! We have replaced Figure 1 with two new figures (new Figures 1 and 2) that are tailored, respectively, to more gently introduce the standard EventProp algorithm and the new extended version with delays.

1. The Yin-Yang dataset, in its original form, only features four input dimensions and does not include a bias signal.

The latter is, unfortunately, often introduced as an additional timing signal and might even be required for some forms of loss functions.

Göltz et al. don't seem to be relying on a bias signal for their benchmarking.

Considering the fact that DelGrad is -- under certain constraints -- very similar in nature to the present work, I would like to see a more direct comparison of the two. For delay-less networks of LIF neurons, EventProp and the formalism behind DelGrad, seem to result in identical gradients.

Including that direct comparison, i.e., training mostly identical networks on identical data, would be potentially interesting to the reader.

Our goal was to compare both the original EventProp publication (bigger networks, 5 input neurons) and DelGrad (smaller networks, 4 input neurons), but we agree that it might have made things confusing. Now we use the same setup as DelGrad, including the loss function, defined at 401-406.

2. I don't fully understand the reasoning behind consistently showing training, validation, and testing accuracy.

It might lead to interesting observations w.r.t. potential overfitting, but also introduces noise.

We changed all our results figures to only show test accuracies.

3. The authors discuss the fact that other studies rely on SHD's test set for model selection and final performance evaluation.

This is generally a very relevant issue, and Hammouamri et al. even clearly state doing so in their manuscript.

I am not sure, however, about the work by Schöne et al.

The authors should clearly point to the methodological issues and trace their claims back to the respective sources ("as XY et al. indicate in their manuscript, they...").

We have added extra information on this at lines 230-235 , indicating the status of each of the mentioned competitor results. However, as Baronig et al. already discuss the methodological issues with SHD training, we did not include a more general discussion of the issue here.

4. I would recommend the authors to use a LaTeX package like `siunitx` to correctly typeset all unit-carrying quantities.

This also prevents a mismatch of significant digits, e.g., in accuracy figures ("98.47±0.004%").

Apologies for the previous inconsistencies. We now use siunitx throughout.

5. In their performance benchmarks, the authors mention the "likely" impact of caching.

I would like to see a less vague explanation, especially considering the fact that the authors also develop the underlying software framework.

Apologies that this was vague, we have expanded this description at lines 300-306.

6. I would like to see the network performance plotted against the number of network parameters.

This likely requires a more extensive sweep across network sizes than what is currently shown in Figures 3, 4, and 5.

Generally, I find the extensive use of bar plots suboptimal, mostly due to their low information density.

Line plots with one line for weight-only and another for delay-augmented networks improve legibility and comparability.

Along those lines, I would also like to see weight-only data for the SHD and SSC datasets.

We have now updated all our results figures according to these recommendations. As per the Nature editorial guidelines, we have had to include the raw data in these plots meaning that a line plot looked very messy so we have instead used error bar plots.

1. Ensure correct and homogeneous typesetting on non-counted/non-variable subscripts.

We updated the typesetting.

2. Mathematical operators, i.e., the differential operator d , are to be set upright. I'd recommend using a suitable LaTeX package for that.

We updated the differential operators.

3. The figures could profit from a bit of love.

This is especially true for Figure 3 B, where the spikes are not clearly visible, and the respective labels seem to be misplaced.

The figure could be generally improved by connecting panels A and B, see, e.g., the original publication on the Yin-Yang dataset.

We now depict the Yin-Yang in a hopefully visually more appealing way.

4. The abstract starts with the theoretical but somewhat vague claim that SNNs are more efficient than ANNs, and inherently better suited than traditional AI methods for time series processing.

We now updated the introduction so that it talks about these issues regarding energy hungry AI in general.

5. The introduction starts similarly generic and vague.

Motivating delay learning on keyword spotting tasks with the energy consumption of training LLMs seems somewhat of a stretch.

This vague and somewhat "populist" introduction does not do the scientific work addressed in this manuscript justice.

We now refer to high energy requirements in general instead of focusing on LLMs specifically.

6. Gradient-based training of SNNs is introduced with the issue of the "non-differentiable Heaviside activation function".

Thresholding in a time-continuous system does not imply the use of a Heaviside function, which is merely a misunderstood implementation detail of time-stepped numerical integration of SNNs.

And the adjoint formalism correctly addresses the underlying issue, as the authors surely know and understand.

This paragraph should be reworked to avoid this common misconception.

That is a very important point, and we do not mention the Heaviside function anymore.

7. The manuscript occasionally relies on colloquial language (e.g., "... into something differentiable.").

We have removed all instances we were able to identify.

8. The authors should more concise when talking about the "exploding memory requirements" and the resulting limits to sequence lengths when using surrogate gradients.

We now refer to this as "memory scales linearly with sequence length".

9. In section 2.4, the authors mention "spike regularizations strength" without ever introducing the concept.

We now introduce it at lines 212-215.

Reviewer #3 (Remarks to the Author):

1) The method is benchmarked only on two standard data sets: SHD and Spiking Speech Commands. On the former, it is very close to or on par with SOTA models. As the authors correctly mention, the performance on this data set has already quite saturated. On the latter, it is clearly below the SOTA for SNNs. Therefore, it is not clear where the advantage of the approach lies. There are several other data sets that could be used for a comparison. The method is not compared in terms of accuracy to other methods for delay learning. Besides the elegance, no measurable advantages are reported.

We have included data points produced using a wider range of previous delay learning techniques in the new versions of figures 5 and 6. We have also added results on a relatively new Braille letter reading dataset, where we outperform previous results. We explored various network sizes with feedforward and recurrent connectivity, and found that with small architectures recurrent networks seem to be significantly better. As no other implementation supports recurrent delays, we believe that our method is the best choice for training small networks.

2) There are many papers on delay learning in SNNs. The authors discuss only a few of them. For example see works cited in their ref [27] and later ones. Performance is only compared to one model in terms of memory usage and training time, where it is relatively clear that the method should have an advantage. But how does it compare to other methods for delay learning such as Slayer or others?

Sadly, many of the previous works on delay learning do not provide code so we are not able to run the type of efficiency benchmark comparisons we made with DCLS. However, we now highlight a wider range of completing delay learning methods in the introduction (lines 79-87), plot performance vs accuracy if the authors evaluated them on SHD or SSC and added some more text (lines 296-300) on how their computational cost and memory usage is likely to compare to our work.

3) The manuscript is very hard to understand if one does not have background knowledge about the adjoint method. The authors could make an attempt to describe their method (and derivation) with more provided intuition and background. Writing could be improved:

- I_p and I_V are not well described. I could not find a definition of the actual functions for the simulations.
- It is not described what t_k stands for - I assume $\{t_k\}$ stands for the set of all spikes in the network. Maybe it would be better to adopt a different notation for that. The set notation looks a bit sloppy to me, in particular in eq. (4) where the delay is also incorporated. It is not well defined how this notation works.
- In Fig. 1 there is a t_1^{\max} which is not defined. Also t^{post} in eq. (10) is not defined.

Apologies that our introduction to EventProp and the adjoint method was previously not clear. We have added introductory text for some intuition (lines 120-138). Instead of our previous introductory figure, we have 2 simpler figures, which hopefully helps with understanding. We updated all of our equations, and hope that now it is easier to follow.

- 4) The figures are below usual standards and hard to parse. For a high class journal, I believe that also nice figures are needed.
- Fig 1: It is very hard to understand from the figure and the description what is happening here. Notation is not well-described (such as d^{hi}_{10} etc)
 - Fig 3: is not acceptable in my opinion (in particular 3B and 3C). Also, the encoding shown in Fig. 3B is not described.
 - Axes labels missing in Fig. 4, 5.

Instead of our previous introductory figure, we have 2 simpler figures, which hopefully helps with understanding. We updated figure 3 so that it is easier to follow. We added axes labels for all dataset examples.

- In 2.2: It seems that the authors are not using their algorithm but rather DelGrad?

Apologies that this wasn't clear but we are using our own algorithm throughout.

- There are some smaller typos (e.g. τ_{mem} and τ_{syn} in the legend of Table 1).

Thank you for spotting these typos. We have corrected all typos that we found.

- lines 44, 45: Surrogate gradients do not 'smooth out' spikes. They assume a smoothed threshold function for the gradient.

We corrected the phrasing.

- Punctuation around equations is often not correct.

We have corrected the punctuation around equations throughout the paper as part of our editing.

Response to reviewers

We thank the reviewers for their thoughtful comments. We have addressed all comments as detailed below and feel that this has helped to improve the manuscript further.

Reviewer #2 (Remarks to the Author):

I am still unsure about Figs. 1 & 2. It might be advisable to invest some work into creating an overview figure (as Fig. 1) to summarize the storyline of the manuscript. This might be more of an editorial decision.

We have considered this suggestion carefully and thought about possible content and design. However, we feel that this paper does not lend itself to the idea of a “graphical abstract” as the concepts that are developed (delay learning in SNNs with the adjoint method) are not graphical in nature and for the rest the story of the paper is very simple: New mathematical derivation -> implementation in mlGeNN -> proof of concept example -> complex examples.

I have one main concern, and that is the performance reported on the Yin-Yang dataset. Göltz et al. report that delays improve performance at iso-parameter count, and in general, the manuscript seems to report significantly lower performance when compared to DelGrad. The authors should either discuss potential reasons for this discrepancy or ensure identical set ups.

We asked the DelGrad authors for their code which allowed us to now follow very closely what they do. Apart from the sigmoid function on the delays (and the fundamental difference of our timestepped internal neuron dynamics implementations), we now have created the same set-up. With this configuration, we now achieve essentially the same results. Within the minor adjustments we made, we found, that allowing multiple spikes in hidden neurons is harmful for this dataset, and that is what made the main difference. The results are updated in the manuscript (Figure 4).

Furthermore, I would recommend to avoid the use of abbreviations, especially in the legends of Figs. 4 to 7.

We have now removed abbreviations in the legends of Figs. 4 to 7.

Lastly, section 2.6 could profit from an improved title. "Computational performance" can refer to both, the computational capabilities of the algorithm and also its cost.

We renamed the section to “Training efficiency”.

Reviewer #3 (Remarks to the Author):

The performance in terms of accuracy is close to the state of the art for SNNs but does not improve beyond that.

This summarises the situation well, but as you say, there are other aspects of the approach that are novel and interesting.

The authors have improved the mathematical description, but could still be improved. For example

It is not mentioned what t_k^{spike} denotes exactly (is $t_1^{\text{spike}}, t_2^{\text{spike}}, \dots$ the (sorted?) list of all spikes in the network?)

We define t_k^{spike} under table 1.

I did not find a definition of $e_n(k)$ in Table 1.

Thank you for calling this to our attention, $e_n(k)$ was an unnecessary term in our formulas, and we removed it.

The notation " t_k^{spike} from i " is sloppy.

To us it felt like a more intuitive way of stating that the sum is over spike times for spikes in neuron i but we have now put a more formal expression that is hopefully clearer and equally well understood.

Are V, I, \dots vectors? This is never explicitly stated.

We now state this under table 1.

The mathematical description in Methods is very technical. The authors did not make an attempt to describe their derivation with more provided intuition and background there.

It is a difficult balance between providing a didactic narrative and keeping things compact and not too repetitive from the existing nice exposition in (Wunderlich and Pehle, 2021). We have edited the Methods one more time to provide some extra intuition on what is happening along the derivation of learning with delays and of delay learning.

In their paper *Efficient Event-based Delay Learning in Spiking Neural Networks*, Mészáros et al. describe an extension to the EventProp formalism, augmenting the latter by trainable synaptic delays. The authors, further, benchmark SNNs with and without delays to demonstrate the advantage of these additional degrees of freedom in terms of classification accuracy. They also compare their approach – in general only relying on sparse information for the backward pass – to the work of Hammouamri, Khalfaoui-Hassani, and Masquelier, highlighting the increase in computational efficiency when relying on sparse gradients.

The authors present an incremental, but yet significant advance in the field of SNN training schemes, and I would be generally inclined to recommend publication of the manuscript (assuming the issues below are addressed). The paper is concisely written and adequately introduces the concept spiking neural networks and the potential power of synaptic transmission delays as additional degrees of freedom. The benchmarking seems mostly sound and I have no significant methodological complaints. Unfortunately, I have neither the background nor the capacity to review the mathematical methods. I fully trust the authors to have correctly derived the delay-aware training adjoint dynamics and their results are backed up and in line with other delay learning approaches, but I can not exclude any issues hidden in the equations. The authors should thus double check their math.

The manuscript could profit from an extended discussion of event-based and dense computation and in that context also differentiate between event-based frameworks – i.e., a mathematical description only relying on computation at the time of (delayed) spike events – and an event-based implementation. This is of particular importance, as the presented implementation is (very likely deliberately) not following that paradigm.

Especially considering the interdisciplinary audience of the journal, I would also like to see a more gentle introduction into the EventProp formalism. The authors already try to hide a lot of the complexity by moving the derivation into the methods section, but confronting the reader with non-trivial math already at the beginning of the results section as well as in Table 1 seems a bit brutal. Considering the fact that the extended formalism is the primary result of the manuscript, I would not recommend to completely hide away the math, but the authors should think of a more friendly way of introducing it. I would, in that context, also recommend to more clearly highlight their own extensions to the delay-less adjoint dynamics, especially in Table 1. Contrasting the impact of delays with the already published state-of-the-art – and assuming that the resulting modifications are straight-forward to highlight – might make that actual delta easier to digest. The same also applies to Figure 1, which is similarly overwhelming. It might be advisable to add a new Figure 1 that gently introduces the main concepts without going into the details.

I enjoyed the fact that the authors benchmarked their formalism for different network topologies and on multiple datasets. However, there are a few minor issues I would like to see addressed:

1. The Yin-Yang dataset, in its original form, only features four input dimensions and does not include a bias signal. The latter is, unfortunately, often introduced as an additional timing signal and might even be required for some forms of loss functions. Göltz et al. don't seem to be relying on a bias signal for their benchmarking. Considering the fact that DelGrad is – under certain constraints – very similar in nature to the present work, I would like to see a more direct comparison of the two. For delay-less networks of LIF neurons, EventProp and the formalism behind DelGrad, seem to result in identical gradients. Including that direct comparison, i.e., training mostly identical networks on identical data, would be potentially interesting to the reader.
2. I don't fully understand the reasoning behind consistently showing training, validation, and testing accuracy. It might lead to interesting observations w.r.t. potential overfitting, but also introduces noise.

3. The authors discuss the fact that other studies rely on SHD’s test set for model selection and final performance evaluation. This is generally a very relevant issue, and Hammouamri, Khalfaoui-Hassani, and Masquelier even clearly state doing so in their manuscript. I am not sure, however, about the work by Schöne et al.. The authors should clearly point to the methodological issues and trace their claims back to the respective sources (“as XY et al. indicate in their manuscript, they...”).
4. I would recommend the authors to use a LaTeX package like `siunitx` to correctly typeset all unit-carrying quantities. This also prevents a mismatch of significant digits, e.g., in accuracy figures (“98.47±0.004%”).
5. In their performance benchmarks, the authors mention the “likely” impact of caching. I would like to see a less vague explanation, especially considering the fact that the authors also develop the underlying software framework.
6. I would like to see the network performance plotted against the number of network parameters. This likely requires a more extensive sweep across network sizes than what is currently shown in Figures 3, 4, and 5. Generally, I find the extensive use of bar plots suboptimal, mostly due to their low information density. Line plots with one line for weight-only and another for delay-augmented networks improve legibility and comparability. Along those lines, I would also like to see weight-only data for the SHD and SSC datasets.

Apart from those points, I would also like the authors to address a number of mostly cosmetic and stylistic issues:

1. Ensure correct and homogeneous typesetting on non-counted/non-variable subscripts.
2. Mathematical operators, i.e., the differential operator d , are to be set upright. I’d recommend using a suitable LaTeX package for that.
3. The figures could profit from a bit of love. This is especially true for Figure 3 B, where the spikes are not clearly visible, and the respective labels seem to be misplaced. The figure could be generally improved by connecting panels A and B, see, e.g., the original publication on the Yin-Yang dataset.
4. The abstract starts with the theoretical but somewhat vague claim that SNNs are more efficient than ANNs, and inherently better suited than traditional AI methods for time series processing.
5. The introduction starts similarly generic and vague. Motivating delay learning on keyword spotting tasks with the energy consumption of training LLMs seems somewhat of a stretch. This vague and somewhat “populist” introduction does not do the scientific work addressed in this manuscript justice.
6. Gradient-based training of SNNs is introduced with the issue of the “non-differentiable Heaviside activation function”. Thresholding in a time-continuous system does not imply the use of a Heaviside function, which is merely a misunderstood implementation detail of time-stepped numerical integration of SNNs. And the adjoint formalism correctly addresses the underlying issue, as the authors surely know and understand. This paragraph should be reworked to avoid this common misconception.
7. The manuscript occasionally relies on colloquial language (e.g., “... into something differentiable.”).
8. The authors should more concise when talking about the “exploding memory requirements” and the resulting limits to sequence lengths when using surrogate gradients.
9. In section 2.4, the authors mention “spike regularizations strength” without ever introducing the concept.

References

- Göltz, Julian, Jimmy Weber, Laura Kriener, Peter Lake, Melika Payvand, and Mihai A Petrovici. 2024. “Delgrad: Exact Gradients in Spiking Networks for Learning Transmission Delays and Weights”. *Arxiv E-Prints*, arXiv-2404.

Hammouamri, Ilyass, Ismail Khalfaoui-Hassani, and Timothée Masquelier. 2023. “Learning Delays in Spiking Neural Networks Using Dilated Convolutions with Learnable Spacings”. *Arxiv Preprint Arxiv:2306.17670*.

Schöne, Mark, Neeraj Mohan Sushma, Jingyue Zhuge, Christian Mayr, Anand Subramoney, and David Kappel. 2024. “Scalable Event-by-Event Processing of Neuromorphic Sensory Signals with Deep State-Space Models”. In *2024 International Conference on Neuromorphic Systems (ICONS)*, 124–31.